# MIXUP MODEL MERGE: ENHANCING MODEL MERGING PERFORMANCE THROUGH RANDOMIZED LINEAR INTERPOLATION

## ABSTRACT

Model merging aims to combine multiple task-specific models into a single model without additional training. Existing merging methods typically apply the same coefficient to all task vectors, which only changes the magnitude of the merged vector while keeping its direction fixed. Since different task-specific models contribute along different directions in parameter space, this restriction can limit the merged model's quality. We propose Mixup Model Merge ($M^3$), a simple method that jointly explores both the merging direction and step size. $M^3$ performs randomized linear interpolation in parameter space, where interpolation coefficients sampled from a chosen distribution determine the direction of the merged task vector, and a scaling factor controls its magnitude. The best merged model is selected using a validation set. Experiments on three task-specific LLMs show that $M^3$ consistently produces higher-quality merged models. Moreover, evaluations on LiveBench and PromptBench demonstrate that $M^3$ substantially improves out-of-distribution and adversarial robustness. $M^3$ also complements sparsification-based methods such as DARE, enabling further performance gains. The code is provided in the supplementary materials.

## 1 INTRODUCTION

In the field of Natural Language Processing (NLP), the emergence of large language models (LLMs) (Brown et al., 2020; Touvron et al., 2023; OpenAI, 2023; Chowdhery et al., 2023) represents a revolutionary breakthrough. With their remarkable capabilities, these models have demonstrated outstanding performance across various tasks (Jiao et al., 2023; Chang et al., 2024b; Nam et al., 2024; Xing, 2024; Guo et al., 2024), significantly advancing NLP technologies.

Supervised Fine-Tuning (SFT) is a crucial technique for adapting LLMs to specific tasks, refining their performance by training on domain-specific data (Hu et al., 2021; Ding et al., 2023; Xia et al., 2024). However, SFT requires substantial computational resources and long training times (Brown et al., 2020; Chang et al., 2024a). To address this challenge, Model Merging has emerged as an efficient solution, fusing the parameters of multiple fine-tuned LLMs into a unified model with diverse capabilities, without the need for additional training or computational costs (Yang et al., 2024; Akiba et al., 2024).

However, existing LLM merging methods typically operate based on task vectors (Ilharco et al., 2022), where a task vector is defined as the directional vector in parameter or feature space corresponding to a task-specific model, representing its task-specific capabilities. In prior approaches, all task vectors are generally assigned the same merging coefficient, which is equivalent to scaling each task vector proportionally (Wortsman et al., 2022; Ilharco et al., 2022; Matena & Raffel, 2022; Jin et al., 2022; Yadav et al., 2023; Yu et al., 2024). In other words, the merged task vector moves along a straight line in the parameter space as the merging coefficient changes, meaning that the coefficient only modifies the magnitude of the merged vector but not its direction.

This limitation can affect the quality of the resulting merged model because the combination directions of different task-specific models' capabilities are not necessarily aligned along a single line. Simple proportional scaling may fail to fully exploit the complementary abilities of the models.

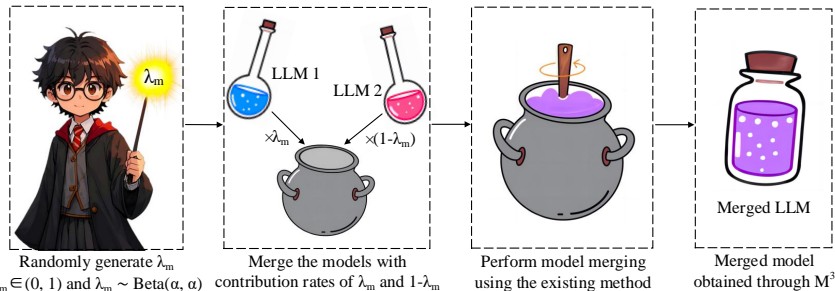

Figure 1: Implementation of M³: A process analogous to proportionally mixing magical potions in Harry Potter. The proposed method controls the contribution ratio between two fine-tuned LLMs by randomly generating a linear interpolation ratio $\lambda_m$, where $\lambda_m \in (0, 1)$ and $\lambda_m \sim \text{Beta}(\alpha, \alpha)$. The distribution of $\lambda_m$ is controlled by adjusting $\alpha$.

This raises a natural question: can assigning different merging coefficients to different task-specific models lead to a better-performing merged model?

To investigate this question, we propose a direction + step-size merging coefficient search framework—Mixup Model Merge (M³), which explores both the merging direction and step size in the task vector space, thereby enabling the discovery of higher-quality merged models in a richer parameter space. Inspired by the randomized linear interpolation strategy in Mixup data augmentation (Zhang, 2017), M³ performs linear interpolation in the model parameter space. The interpolation coefficients are sampled from a specific distribution to determine the merging direction (i.e., the direction of the merged model's task vector), and the scaling step size is then adjusted to control the magnitude of the merged task vector. Finally, the best-performing merged model is selected on a held-out validation set.

We conducted extensive experiments with three homologous task-specific fine-tuned LLMs: WizardLM-13B (Xu et al., 2024), WizardMath-13B (Luo et al., 2023), and llama-2-13b-code-alpaca (Chaudhary, 2023), which specialize in instruction following, mathematical reasoning, and code generation, respectively. Inspired by Mixup's effectiveness in enhancing the robustness of neural networks when handling corrupted labels or adversarial examples (Zhang, 2017), we further performed comprehensive evaluations on LiveBench (White et al., 2024) and PromptBench (Zhu et al., 2024) to validate the potential of M³ in improving the OOD robustness and adversarial robustness (Wang et al., 2023; Zhu et al., 2023) of merged models. The experimental results demonstrate that our proposed M³ method can significantly improve merged models' performance across various tasks (as shown in Figure 2a), and enhance the OOD and adversarial robustness of the merged models (as shown in Figure 2b).

## 2 RELATED WORKS

### 2.1 MODEL MERGING

Model merging is a technique that integrates the parameters of multiple models to create a unified model with enhanced or diverse capabilities (Wortsman et al., 2022; Ilharco et al., 2022; Matena & Raffel, 2022; Jin et al., 2022; Yadav et al., 2023; Yu et al., 2024; Lin et al., 2024). Task arithmetic (Ilharco et al., 2022) leverages task vectors for model merging through arithmetic operations, incorporating a predefined scaling term to weight the contribution of different models. Fisher Merging (Matena & Raffel, 2022) performs parameter fusion by applying weights derived from the Fisher information matrix (Fisher, 1922), resulting in more precise parameter integration. TIES-Merging (Yadav et al., 2023) addresses task conflicts by removing low-magnitude parameters, resolving sign disagreements, and merging only the parameters that align with the final agreed-upon sign. In (Yu et al., 2024), it is found that LLMs can enhance their capabilities through model merging. Additionally, it introduces DARE, a method for sparsifying the delta parameters of the model (Ilharco et al., 2022), significantly improving the performance of various model merging techniques. DELLA (Deep et al., 2024) is a novel model merging technique that integrates a new pruning strategy called

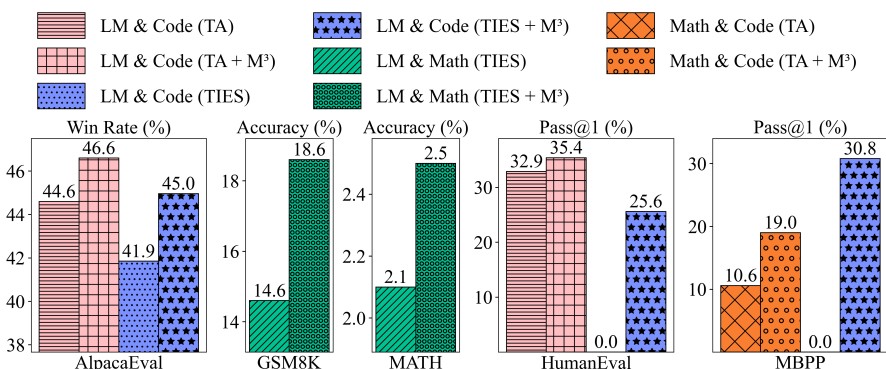

(a) Performance of the merged models obtained through Task Arithmetic and TIES-Merging with or without $M^3$.

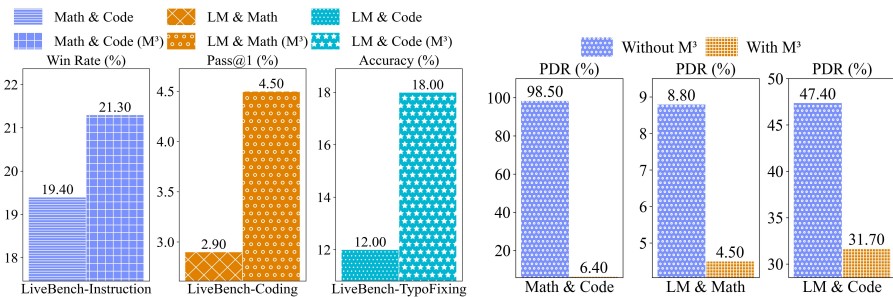

(b) (Left) Performance on OOD datasets. (Right) Adversarial robustness (PDR) of merged models with or without $M^3$.

Figure 2: (a) $M^3$ significantly improves the performance of model merging. (b) OOD and adversarial robustness of merged models with or without $M^3$.

MAGPRUNE, which samples delta parameters based on their magnitudes. MAGPRUNE demonstrates improvements over existing methods such as DARE and TIES.

## 3 METHODOLOGY

For each task $t$, define the task vector as

$$\delta_t = \theta^t_{\text{SFT}} - \theta_{\text{PRE}} \in \mathbb{R}^d,$$

For each task $t$, define the task vector as $\delta_t = \theta^t_{\text{SFT}} - \theta_{\text{PRE}} \in \mathbb{R}^d$, where $\theta_{\text{PRE}} \in \mathbb{R}^d$ denotes the parameters of the pretrained model, $\theta^t_{\text{SFT}} \in \mathbb{R}^d$ denotes the parameters of the model fine-tuned on task $t$, and $d$ is the total number of parameters. The task vector $\delta_t$ represents the parameter-space change induced by task-specific fine-tuning.

Geometrically, each task vector can be decomposed into a direction and a magnitude:

$$u_t = \frac{\delta_t}{\|\delta_t\|}, \quad s_t = \|\delta_t\|,$$

where $u_t$ is a unit vector indicating the direction of task-specific adaptation, and $s_t$ is the magnitude of the update, reflecting the relative impact of the task on the model parameters. Misaligned directions indicate potential task conflicts, while large magnitudes may dominate merges.

$M^3$ performs interpolation in this task-vector space. For two tasks, the merged task vector is

$$\delta_M = \lambda \, \delta_{t_1} + (1 - \lambda) \, \delta_{t_2}, \quad \lambda \in (0, 1), \quad \lambda \sim P(\phi),$$

and for $K$ tasks, it generalizes to

$$\delta_M = \sum_{k=1}^{K} \lambda_k \, \delta_{t_k}, \quad \boldsymbol{\lambda} \sim P(\boldsymbol{\phi}),$$

where $\lambda$ or $\lambda_k$ control the relative contribution of each task-specific vector, $P$ denotes a family of distributions, and $\phi$ are distribution parameters controlling the sampling (e.g., shape, scale).

The final merged model is then

$$\theta_M = \theta_{\text{PRE}} + s \, \delta_M,$$

where $s$ is a scaling term that adjusts the overall magnitude of the merged task vector.

In summary, $M^3$ explicitly separates *direction* and *magnitude* of task vectors, and controls the merged model by adjusting both the interpolation coefficients $\lambda$ (affecting direction) and the scaling factor $s$ (affecting magnitude). This provides a unified framework for multi-task model merging.

### 3.1 Mixup Model Merge for Two-Model Merging

We consider two task-specific fine-tuned language models, where the interpolation coefficient $\lambda$ is randomly sampled from a Beta distribution, i.e., $\lambda \sim$ Beta$(\alpha, \alpha)$. The Beta distribution is defined over the interval $(0, 1)$, making it naturally suitable for interpolation and introducing controllable randomness into the model merging process. Its tunable shape parameter $\alpha$ allows us to adjust the sampling bias of $\lambda$. Specifically, as illustrated in Figure 3: (1) When $\alpha < 1$, the distribution of $\lambda$ is bimodal, with higher probabilities near the extremes (0 and 1), indicating that the merged model is more likely to be dominated by one of the two task-specific models. As $\alpha$ decreases, the range of sampled $\lambda$ values becomes more concentrated, improving exploration efficiency but reducing the diversity of merged models explored. (2) When $\alpha > 1$, the distribution of $\lambda$ is concentrated around the middle region (e.g., near 0.5), resulting in more balanced contributions from both models. As $\alpha$ increases, the range of sampled $\lambda$ values also becomes more concentrated, enhancing exploration efficiency while reducing the diversity of merged models. (3) When $\alpha = 1$, $\lambda$ follows a uniform distribution, meaning that all values within the interval $(0, 1)$ are equally likely to be sampled. As $\alpha$ approaches 1, the distribution of $\lambda$ becomes more dispersed, leading to lower exploration efficiency but higher diversity of the merged models explored.

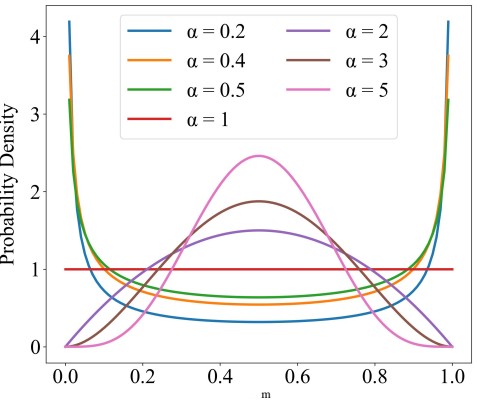

Figure 3: The Beta distribution visualization for different $\alpha$ values.

### 3.2 Empirical Observation: Asymmetry in Optimal Merge Ratios

Across extensive experiments on multiple two-model combinations (e.g., LM, Math, Code) and three representative merging paradigms — Task Arithmetic, TIES-Merging, and DARE — we repeatedly observe a highly consistent phenomenon: The best-performing merged models (1) achieve balanced or even improved task performance compared to individual fine-tuned experts, and (2) do not significantly degrade any expert task. Importantly, these optima almost always occur at highly uneven interpolation ratios, where one task-specific model dominates the merge.

This empirical pattern suggests a clear inductive bias: During model merging, the contributions of different task-specific models should be asymmetric. Some models require substantial weight, while others must be significantly down-weighted to avoid conflicts between task vectors.

### 3.3 P-Series-based Mixup Model Merging for Multiple Models

For merging $K > 2$ task-specific models, we extend $M^3$ with a $p$-series weight allocation. The merged task vector is

$$\delta_M = \sum_{k=1}^{K} \lambda_k \, \delta_{t_k}, \quad \lambda_k = \frac{k^{-p}}{\sum_{j=1}^{K} j^{-p}}, \quad \delta_M \leftarrow s \, \delta_M,$$

where $\lambda_k$ determines the relative contribution (direction) of each task vector, following a monotonically decreasing $p$-series (dominant model first), and $s$ is a global scaling factor controlling the overall magnitude of the merged update.

The merged model is then

$$\theta_M = \theta_{\text{PRE}} + \delta_M.$$

This formulation extends the two-model $M^3$ interpolation to multiple tasks. The $p$-series produces an asymmetric weighting of task vectors, reflecting the empirical observation that optimal merges are usually dominated by one or a few tasks. By separating direction (given by $\lambda_k$) and magnitude ($s$), we can flexibly explore low-loss merging points and achieve balanced performance across multiple tasks.

### 3.4 Theoretical Analysis

**Feasibility of Interpolation.** We perform direct parameter interpolation among multiple task-specific models because models fine-tuned from the same pretrained model typically reside in a shared, flat, and connected low-loss region Neyshabur et al. (2020). Therefore, linear interpolation between these models generally remains within this low-loss region, avoiding significant performance degradation. In some cases, the interpolated model may even outperform the original endpoint models.

**Mechanism of Performance Improvement: Mitigating Parameter Conflicts.** Let $\theta_{\text{PRE}}$ denote the pretrained model, and let $N$ task-specific models be obtained by fine-tuning it: $\theta_{\text{SFT}}^{t_1}, \theta_{\text{SFT}}^{t_2}, \ldots, \theta_{\text{SFT}}^{t_N}$. The task vectors are defined as:

$$\delta_{t_k} = \theta_{\text{SFT}}^{t_k} - \theta_{\text{PRE}}, \quad k = 1, \ldots, N.$$

The $M^3$ merged model is then defined as:

$$\theta_M = \theta_{\text{PRE}} + s \sum_{k=1}^{N} \lambda_k \, \delta_{t_k}, \quad \lambda_k \geq 0, \quad \sum_{k=1}^{N} \lambda_k = 1,$$

where $s > 0$ is a global scaling factor controlling the overall magnitude, and $\lambda_k$ controls the relative contribution (direction) of each task-specific model.

When some components of the task vectors $\delta_{t_k}$ have opposing signs (i.e., conflicts in certain directions), choosing suitable interpolation coefficients $\{\lambda_k\}$ can partially cancel these conflicting components:

$$\sum_{k=1}^{N} \lambda_k \, \delta_{t_k}^{(i)} \approx 0, \quad i = 1, \ldots, d.$$

This suppresses conflicting updates and improves stability. Compared to using the same coefficient for all models, multi-coefficient interpolation allows flexible adjustment of each task-specific model's contribution, better leveraging their complementary abilities.

**Achieving High-Quality Merges via Interpolation.** Through this conflict mitigation mechanism, different configurations of $\{\lambda_k\}$ correspond to different merging directions in parameter space. By searching over both direction (via $\lambda_k$) and step size (via $s$), we can identify configurations that maximize conflict suppression while balancing task-specific abilities, resulting in higher-quality merged models.

| Merging Methods | Models | Use $M^3$ (Ours) | Sampling Details | | Instruction Following | Mathematical Reasoning | | Code Generating | | Avg. |
|---|---|---|---|---|---|---|---|---|---|---|
| | | | $\lambda_m$ | $\alpha$ | AlpacaEval | GSM8K | MATH | HumanEval | MBPP | |
| - | LM | - | - | - | 45.14 | 2.20 | 0.04 | 36.59 | 34.00 | - |
| | Math | - | - | - | - | 64.22 | 14.02 | - | - | - |
| | Code | - | - | - | - | - | - | 23.78 | 27.60 | - |
| Task Arithmetic | LM | No | - | - | 45.78 | 66.34 | 13.40 | - | - | 41.84 |
| | & Math | Yes | 0.34 | 2 | 41.65 | 66.34 | 13.74 | - | - | 40.58 |
| | LM | No | - | - | 44.64 | - | - | 32.93 | 33.60 | 37.06 |
| | & Code | Yes | 0.99 | 0.4 | 46.64 | - | - | 35.37 | 33.80 | 38.60 |
| | Math | No | - | - | - | 64.67 | 13.98 | 8.54 | 8.60 | 23.95 |
| | & Code | Yes | 0.98 | 0.5 | - | 63.53 | 13.94 | 7.93 | 19.00↑ | 26.1 |
| TIES-Merging | LM | No | - | - | 38.63 | 14.56 | 2.12 | - | - | 18.44 |
| | & Math | Yes | 0.62 | 1 | 38.73 | 18.57↑ | 2.48 | - | - | 19.92 |
| | LM | No | - | - | 41.85 | - | - | 0.0 | 0.0 | 13.95 |
| | & Code | Yes | 0.84 | 0.5 | 44.96 | - | - | 25.61↑ | 30.80↑ | 33.79↑ |
| | Math | No | - | - | - | 64.67 | 13.68 | 9.15 | 22.60 | 27.57 |
| | & Code | Yes | 0.63 | 3 | - | 64.75 | 14.16 | 9.76 | 21.4 | 27.52 |
| DARE | LM | No | - | - | **49.00** | 66.64 | 13.2 | - | - | 42.95 |
| | & Math | Yes | 0.38 | 0.4 | 44.90 | **67.32** | 13.74 | - | - | 41.98 |
| | LM | No | - | - | 41.47 | - | - | 35.98 | 33.00 | 36.82 |
| | & Code | Yes | 0.99 | 0.5 | 45.20↑ | - | - | 35.98 | **35.20** | 38.79 |
| | Math | No | - | - | - | 65.05 | 10.37 | 9.15 | 9.80 | 23.59 |
| | & Code | Yes | 0.98 | 0.5 | - | 65.13 | **14.32**↑ | 8.54 | 18.00↑ | 26.50↑ |

Table 1: Comparison of Merged Model Performance Before and After Applying $M^3$. We use three task-specific LLMs: WizardLM-13B (LM), WizardMath-13B (Math), and llama-2-13b-codealpaca (Code). Bold indicates the best results for each dataset, underline highlights improvements with $M^3$, and ↑ denotes significant gains after introducing $M^3$.

# 4 EXPERIMENTS

## 4.1 EXPERIMENTAL SETUP

**Task-Specific Fine-Tuned LLMs and Datasets.** Following the experimental setup given in Yu et al. (2024), we select three task-specific fine-tuned LLMs: WizardLM-13B (Xu et al., 2024), WizardMath-13B (Luo et al., 2023), and llama-2-13b-code-alpaca (Chaudhary, 2023), all of which use Llama-2-13b (Touvron et al., 2023) as the pre-trained backbone. These models are respectively designed for instruction-following, mathematical reasoning, and code generation tasks. To evaluate the instruction-following task we use AlpacaEval (Li et al., 2023). For testing mathematical reasoning task, we employ GSM8K (Cobbe et al., 2021) and MATH (Hendrycks et al., 2021). For estimating the performance of code-generating task, we use HumanEval (Chen et al., 2021) and MBPP (Austin et al., 2021). More details of these LLMs and datasets can be found in Appendix A.

**Benchmarks for evaluating Out-of-Distribution and Adversarial Robustness.** To evaluate OOD robustness, we evaluate merged models including math & code, LM & math, and LM & code on instruction following (LiveBench-Instruction), coding (LiveBench-Coding), and language comprehension (LiveBench-TypoFixing) category in LiveBench (White et al., 2024), respectively. More details on OOD benchmarks are given in Appendix B.

We utilize the Adversarial Prompt Attacks module in PromptBench (Zhu et al., 2024) to evaluate the robustness of LLMs against adversarial prompts. Specifically, we employ three attack methods: DeepWordBug (character-level) (Gao et al., 2018), BERTAttack (word-level) (Li et al., 2020), and StressTest (sentence-level) (Naik et al., 2018). The evaluation is conducted on two datasets supported by PromptBench: SST2 (sentiment analysis) (Socher et al., 2013) and CoLA (grammatical correctness) (Warstadt, 2019). For more details on PromptBench and attack methods, please refer to Appendix C.

**Evaluation Metrics.** We calculate win rate for AlpacaEval and LiveBench-Instruction, zero-shot accuracy for GSM8K and MATH, pass@1 for HumanEval, MBPP and LiveBench-Coding, Matthews correlation coefficient (MCC) for CoLA, accuracy for SST2, and zero-shot accuracy for LiveBench-TypoFixing.

| Model | Dataset | Use Mixup | Use Attack | Metric (%) | PDR (%) |
|---|---|---|---|---|---|
| Math & Code | SST2 | No | No | 57.68 | 38.97 |
| | | | Yes | 35.21 | |
| | | Yes | No | 86.24 | **35.77** |
| | | | Yes | 55.39 | |
| | CoLA | No | No | 45.54 | 98.53 |
| | | | Yes | 0.67 | |
| | | Yes | No | 71.72 | **6.42** |
| | | | Yes | 67.11 | |
| LM & Math | SST2 | No | No | 92.78 | **29.05** |
| | | | Yes | 65.83 | |
| | | Yes | No | 91.28 | 34.55 |
| | | | Yes | 59.75 | |
| | CoLA | No | No | 79.19 | 8.84 |
| | | | Yes | 72.20 | |
| | | Yes | No | 80.54 | **4.52** |
| | | | Yes | 76.89 | |
| LM & Code | SST2 | No | No | 10.55 | 38.04 |
| | | | Yes | 6.54 | |
| | | Yes | No | 73.17 | **7.68** |
| | | | Yes | 67.55 | |
| | CoLA | No | No | 74.21 | 47.42 |
| | | | Yes | 39.02 | |
| | | Yes | No | 74.78 | **31.67** |
| | | | Yes | 51.10 | |

Table 2: Adversarial robustness of merged models on SST2 and CoLA with StressTest prompt attacks. Best and second-best results are marked in bold and underlined, respectively.

**Implementation Details.** In each distinct merging experiment—where "distinct" refers to variations in either the task-specific LLMs or the model merging methods—we perform a full sweep over the sole hyperparameter of $M^3$, $\alpha$, using the set $\{0.2, 0.4, 0.5, 1, 2, 3, 5\}$. For each $\alpha$ value, we sample $\lambda_m$ once from the corresponding Beta distribution, resulting in a total of seven samples in each merging experiment. Seven samples are not the minimum required to observe significant performance gains. In most cases, a smaller number of samples is sufficient to identify a performant interpolation coefficient. However, to enable a more comprehensive exploration, we fix the number of samples to seven—each corresponding to a representative shape of the Beta distribution. While the stochastic nature of our method means there may be cases where none of the seven samples yields a significant improvement, such cases are not often observed in our experiments and thus not considered. Given the strong performance observed with just seven samples, and to conserve computational resources, we refrain from further sampling.

Unless otherwise specified, the remaining details of the model merging experiments follow Yu et al. (2024). For a detailed description of the hyperparameter settings used in the representative existing model merging methods, please refer to Appendix D. All experiments are conducted on NVIDIA GeForce RTX 4090 GPUs.

## 4.2 MERGING TWO TASK-SPECIFIC FINE-TUNED LLMS

Our proposed method, $M^3$, is a plug-and-play approach. To evaluate its effectiveness in improving merged model performance, we integrate $M^3$ into several representative existing model merging methods (Average Merging, Task Arithmetic, TIES-Merging, DARE) and compare the performance of the merged models before and after applying $M^3$. The results is presented in Table 1.

**Performance Gains of $M^3$ Across Merging Methods and Tasks.** As shown in Table 1, $M^3$ generally enhances existing model merging methods, achieving over 10-point improvements on multiple datasets and up to 30 points on one. For example, the improvements achieved by Average Merging with $M^3$ for Math & Code are 7.43% on GSM8K, 3.74% on Math, and 11.0% on MBPP. For LM & Code, Average Merging with $M^3$ shows improvements of 7.31% on AlpacaEval, 7.32% on HumanEval, and 2.4% on MBPP. Task Arithmetic with $M^3$ results in improvements of 2.0% on AlpacaEval and 2.44% on HumanEval for LM & Code, and 10.4% on MBPP for Math & Code. TIES-Merging with $M^3$ achieves an improvement of 4.01% for LM & Math on GSM8K. For LM & Code, TIES-Merging with $M^3$ shows significant improvements of 3.11% on AlpacaEval, 25.61%

on HumanEval, and 30.8% on MBPP. Furthermore, as evidenced by the Avg. column in Table 1, $M^3$ consistently improves the overall performance of the merged models. This enhancement is particularly crucial for scenarios where the merged models must strike a balance between task-specific capabilities. A few datasets do not show performance improvements with $M^3$, but the results remain on par with those without $M^3$, indicating that it does not degrade overall performance. Moreover, such cases are negligible compared to the overall benefits brought by $M^3$.

As shown in Table 1, $M^3$ achieves the best performance on four out of five task-specific datasets, surpassing even the individually fine-tuned LLMs (i.e., LM, Math, and Code). This suggests that $M^3$ enables the discovery of interpolated models along linear paths between two models that outperform both endpoints, providing support for the theoretical insights discussed in Section 3.4.

In addition, we also compare our proposed method $M^3$ with existing model merging methods. The details are in Appendix E.

**Mitigating Suboptimal Fine-Tuning with $M^3$.** Yu et al. (2024) noted that the poor performance of merging WizardMath-13B with llama-2-13b-code-alpaca stems from the latter not being sufficiently fine-tuned for code generation tasks. As shown in Table 1, the proposed $M^3$ method improves the pass@1 score on the code generation dataset MBPP by 10.4%. This improvement demonstrates that even when one of the fine-tuned models is suboptimal for a specific task, $M^3$ can effectively unlock the merging potential by adaptively exploring suitable interpolation coefficients (i.e., contribution ratios), thereby maximizing the performance of the merged model. In other words, $M^3$ significantly mitigates the negative impact of insufficient fine-tuning on merging quality, further highlighting the critical importance of task-specific LLMs' contribution ratio in determining the performance of the merged model.

### 4.3 MERGING THREE TASK-SPECIFIC FINE-TUNED LLMS

We conducted systematic experiments on **three task-specific models** (LM, Math, and Code). (1) As shown in Table 3, the results indicate that the trade-offs among the three tasks become **more balanced**, leading to improved performance for the Task Arithmetic merging method. (2) Moreover, Table 4 demonstrates that we can effectively guide the three-model merge based on the optimal merge ratios observed in pairwise model merges. Specifically, for two-model merges, the optimal ratios follow the order Math > Code, Math > Instruct, and Instruct > Code. Accordingly, for the three-model merge, we set the weights in the order Math > Instruct > Code. Experiments show that this weight ordering clearly directs the merge and outperforms other possible weight configurations. (3) As shown in Table 5, when the model merging method is held constant, the performance of the merged models varies with different values of $p$ and $s$, generally showing an initial increase followed by a decrease. This demonstrates the effectiveness of our approach in optimizing both the merge direction and step size.

| Merge method | AlpacaEval | GSM8K | MATH | HumanEval | MBPP |
|---|---|---|---|---|---|
| baseline (Task Arithmetic + scaling term) | 6.27 | 58.52 | 9.86 | 18.29 | 29.40 |
| Ours | 7.20 | 67.55 | 14.56 | 17.68 | 25.80 |

Table 3: Performance of three merged models instruct, math, and code.

| p | s | Merge Weights | AlpacaEval | GSM8K | MATH | HumanEval | MBPP |
|---|---|---|---|---|---|---|---|
| 1 | 1 | 0.5455 math + 0.2727 instruct + 0.1818 code | 4.60 | 63.08 | 12.66 | 23.17 | 30.40 |
| 1 | 1 | 0.7347 instruct + 0.1837 math + 0.0816 code | 10.37 | 2.650 | 0.12 | 34.70 | 35.80 |
| 1 | 1 | 0.7347 math + 0.1837 code + 0.0816 instruct | 4.22 | 62.90 | 13.56 | 8.53 | 24.40 |

Table 4: Performance of different weight orderings.

| p | s | Merge Weights | AlpacaEval | GSM8K | MATH | HumanEval | MBPP |
|---|---|---|---|---|---|---|---|
| 3 | 1 | 0.8607 math + 0.1076 instruct + 0.0319 code | 3.73 | 66.72 | 14.42 | 9.760 | 22.40 |
| 2 | 1 | 0.7347 math + 0.1837 instruct + 0.0816 code | 3.98 | 65.35 | 13.20 | 10.98 | 25.80 |
| 1 | 1 | 0.5455 math + 0.2727 instruct + 0.1818 code | 4.60 | 63.08 | 12.66 | 23.17 | 30.40 |
| 0 | 1 | 0.3333 math + 0.3333 instruct + 0.3333 code | 6.27 | 58.53 | 9.860 | 18.29 | 29.40 |
| 1 | 1.7 | 1.7*(0.5455 math + 0.2727 instruct + 0.1818 code) | 7.20 | 67.55 | 14.56 | 17.68 | 25.80 |
| 1 | 2.0 | 2.0*(0.5455 math + 0.2727 instruct + 0.1818 code) | 6.83 | 65.13 | 12.84 | 13.41 | 23.60 |

Table 5: Performance of model merging with varying p and step size. In the paper, we will update this part to show the trends using line plots.

## 4.4 MODEL ROBUSTNESS

**Out-of-Distribution Robustness.** To ensure the evaluation datasets reflect true OOD scenarios, we select recently released datasets from domains not seen during fine-tuning. Accordingly, Math & Code is evaluated on LiveBench-Instruction, LM & Math on LiveBench-Coding, and LM & Code on LiveBench-TypoFixing. As shown in Figure 4, $M^3$ consistently improves OOD performance across all model pairs and merging methods. For example, under Task Arithmetic, $M^3$ yields gains of 1.9%, 1.6%, and 6% on the three respective tasks. Similar improvements are observed with Average Merging (1.5%, 0.7%, 4%) and TIES-Merging (1.1%, 0.6%, 14%). These results highlight $M^3$'s effectiveness in enhancing OOD generalization.

**Adversarial robustness.** We evaluate the adversarial robustness of three merged models (Math & Code, LM & Math, and LM & Code) using three prompt attack methods from PromptBench (Zhu et al., 2024): DeepWordBug, BERTAttack, and StressTest. For DeepWordBug and BERTAttack, we randomly select three word positions per prompt. Robustness is measured by Performance Drop Rate (PDR) (Zhu et al., 2023), with lower PDR indicating stronger robustness (see Appendix F for details). As shown in Table 2, $M^3$ notably improves robustness under StressTest attacks. For instance, it reduces PDR by 3.2% (SST2) and 92.12% (CoLA) for Math & Code, and by 30.36% (SST2) and 15.75% (CoLA) for LM & Code. Additionally, $M^3$ improves performance metrics such as accuracy and MCC—e.g., accuracy for LM & Code on SST2 increases by 62.62%. These results indicate that $M^3$ enhances both robustness and task performance. Results for DeepWordBug and BERTAttack are provided in Appendix G.

A possible explanation for the observed improvements is that parameter updates from different tasks may conflict along certain directions in parameter space, making merged models sensitive to perturbations or out-of-distribution inputs. By adjusting the merging direction and interpolating parameters with carefully chosen coefficients, $M^3$ can alleviate such conflicts, effectively balancing contributions from different tasks. This mechanism likely underlies the enhanced robustness and sustained task performance observed under both adversarial and OOD conditions.

## 4.5 COMPARISON OF MIXUP MODEL MERGE AND DARE

DARE is a model sparsification method proposed by Yu et al. (2024) (see Appendix H for details). We compare $M^3$ and DARE by integrating both into three representative model merging methods: Average Merging, Task Arithmetic, and TIES-Merging (full results in Appendix I). The drop rate for DARE is fixed at 0.2. As shown in Table 11 in appendix, $M^3$ generally outperforms DARE and achieves significantly better results on several datasets. For example, on MBPP, the pass@1 score of the Math & Code model increases from 9.8% with DARE to 19% with $M^3$. Moreover, combining $M^3$ with DARE typically leads to even stronger performance, suggesting they are complementary.

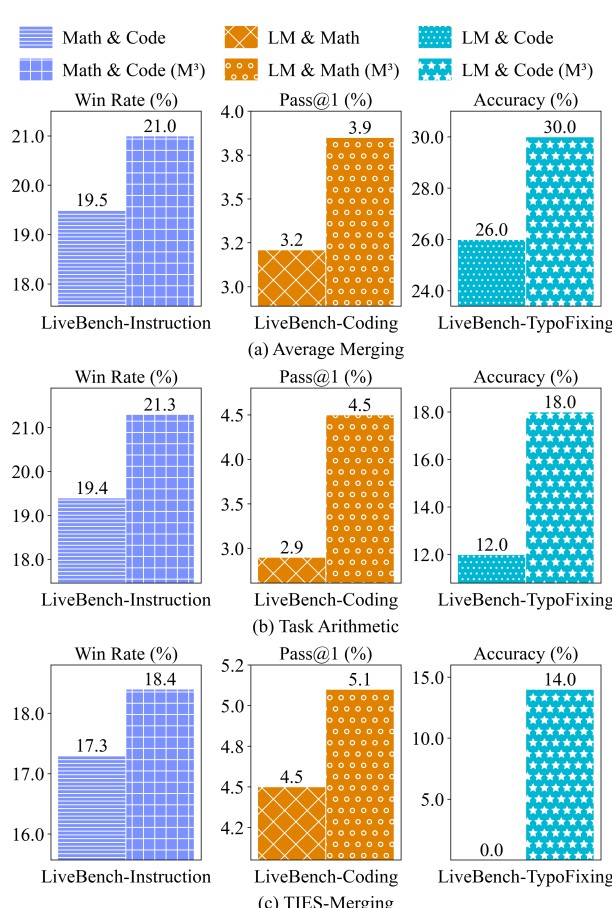

Figure 4: Performance of merged models (Math & Code, LM & Math, and LM & Code) using three model merging methods (Average Merging, Task Arithmetic, and TIES-Merging) on OOD datasets.

Notably, $M^3$ alone often yields the best results, while DARE alone rarely does, further highlighting the effectiveness of $M^3$.

## 5 CONCLUSION

Existing model merging methods often assign the same coefficient to all task vectors, limiting the merged model to a fixed direction and underutilizing complementary capabilities. We propose Mixup Model Merge ($M^3$), which jointly explores merging direction and step size via randomized linear interpolation in parameter space. Interpolation coefficients sampled from a distribution (e.g., Beta) allow flexible exploration of contribution ratios, and the best merged model is selected on a validation set. Experiments on three task-specific LLMs show that $M^3$ consistently improves task performance and enhances out-of-distribution and adversarial robustness. $M^3$ also complements sparsification methods such as DARE. This simple, effective approach provides a general framework for optimizing model merging and can potentially extend to merging with RLHF-aligned models to reduce alignment costs.

## 6 ETHICS STATEMENT

This research does not involve human subjects, personally identifiable information, or sensitive data; therefore, no Institutional Review Board (IRB) approval was required. All datasets used are publicly

available and released under appropriate licenses. Our work poses no apparent ethical risks and has no conflicts of interest.

## 7 REPRODUCIBILITY STATEMENT

To ensure reproducibility, we provide detailed descriptions of the models and experimental settings in the main text, with additional hyper-parameter configurations and implementation details included in the appendix. All datasets used in our experiments are publicly available. Furthermore, we provide the source code, configuration files, and execution scripts in the supplementary material, enabling other researchers to faithfully reproduce our results.

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

## A    TASK-SPECIFIC FINE-TUNED LLMS AND DATASETS DETAILS

We conduct model merging experiments using three task-specific LLMs fine-tuned from Llama-2-13b:

- **WizardLM-13B** is an instruction-following model based on Llama-2-13b, designed to improve open-domain instruction-following. Using the Evol-Instruct method (Xu et al., 2024), it generates high-complexity instruction data to reduce human annotation and enhance generalization. The model undergoes supervised fine-tuning with AI-generated data, followed by refinement via RLHF. Evaluation results show that Evol-Instruct-generated instructions outperform human-written ones, and WizardLM-13B surpasses ChatGPT in high-complexity tasks. In GPT-4 automated evaluation, it achieves over 90% of ChatGPT's performance in 17 out of 29 tasks, demonstrating the effectiveness of AI-evolved instruction fine-tuning (Xu et al., 2024).

- **WizardMath-13B**, optimized from Llama-2-13b, is designed for mathematical reasoning and enhances Chain-of-Thought (CoT) (Wei et al., 2022) capabilities. It uses Reinforcement Learning from Evol-Instruct Feedback to evolve math tasks and improve reasoning. Trained on GSM8K and MATH datasets, it excels in both basic and advanced math problems. In evaluations, WizardMath-Mistral 7B outperforms all open-source models with fewer training data, while WizardMath 70B surpasses GPT-3.5-Turbo, Claude 2, and even early GPT-4 versions in mathematical reasoning tasks.

- **llama-2-13b-code-alpaca** is a code generation model fine-tuned from Llama-2-13b, designed to enhance code understanding and generation. It follows the same training approach as Stanford Alpaca (Taori et al., 2023) but focuses on code-related tasks. The model is fine-tuned with 20K instruction-following code samples generated using the Self-Instruct method (Wang et al., 2022). However, as it has not undergone safety fine-tuning, caution is required when using it in production environments.

We use one dataset to evaluate the instruction-following task:

- **AlpacaEval** (Li et al., 2023) is an LLM-based automated evaluation metric that assesses model performance by testing on a fixed set of 805 instructions and computing the win rate of the evaluated model against a baseline. The evaluation process involves an LLM-based evaluator that compares the responses and determines the probability of preferring the evaluated model. In this paper, we use AlpacaEval 2.0 (Dubois et al., 2024). To reduce costs, we use chatgpt_fn for evaluation.

We use two dataset to evaluate the mathematical reasoning task:

- **GSM8K** is a dataset of 8.5K high-quality, linguistically diverse grade school math word problems, designed to evaluate the multi-step mathematical reasoning abilities of large language models. It consists of 7.5K training problems and 1K test problems. In this paper, we use the 1K test set for evaluation (Cobbe et al., 2021).

- **MATH** is a dataset containing 12,500 competition-level math problems, designed to evaluate and enhance the problem-solving abilities of machine learning models. It consists of 7,500 training problems and 5,000 test problems. We use the 5,000 test set for evaluation (Hendrycks et al., 2021).

We used two dataset to evaluate the code generation task:

- **HumanEval** is a dataset consisting of 164 hand-written programming problems, designed to evaluate the functional correctness of code generation models. Each problem includes a function signature, docstring, function body, and unit tests. The dataset tests models' language comprehension, reasoning, and algorithmic abilities (Chen et al., 2021).

- **MBPP** is a dataset containing 974 programming problems designed to evaluate a model's ability to synthesize Python programs from natural language descriptions. The problems range from basic numerical operations to more complex tasks involving list and string processing. The test set consists of 500 problems, which are used for evaluation in this paper (Austin et al., 2021).

## B  OUT-OF-DISTRIBUTION DATASET SELECTION DETAILS

LiveBench (White et al., 2024) is a dynamic benchmark for large language models, featuring frequently updated questions and diverse tasks. To assess OOD robustness, we evaluate math & code, LM & math, and LM & code models using instruction following (LiveBench-Instruction), coding (LiveBench-Coding), and language comprehension (LiveBench-TypoFixing) category in LiveBench, respectively, deliberately avoiding the fine-tuning domains of the merged fine-tuned models. These tasks were released after November 2023, whereas WizardLM-13B, WizardMath-13B, and llama-2-13b-code-alpaca were all introduced earlier. Furthermore, their shared Llama-2-13b backbone was trained on data only up to July 2023. Consequently, these factors collectively ensure that the evaluation remains OOD in the temporal dimension.

When assessing the OOD robustness of LM & Code using the Language Comprehension category in LiveBench, only the typo-fixing task is considered. This decision is based on the fact that LiveBench is highly challenging, and the merged model performs poorly on other tasks in this category, with accuracy close to zero, rendering the evaluation results inconclusive and uninformative.

Finally, we acknowledge the limitations of these datasets. For large models like Llama-2-13b, identifying truly OOD datasets is difficult, as their training data likely covers similar distributions. These datasets are better described as "out-of-example", representing instances not explicitly seen during training. As discussed in (Wang et al., 2023), distribution shifts can occur across domains and time. While Llama-2-13b may have been trained on datasets for tasks like instruction-following, coding, and language comprehension, the datasets we selected remain valuable for OOD evaluation by capturing temporal shifts, providing insights into robustness over time.

## C  ADVERSARIAL ROBUSTNESS EVALUATION EXPERIMENTS SETTING DETAILS

PromptBench (Zhu et al., 2024) is a unified library designed for evaluating LLMs, providing a standardized and extensible framework. It includes several key components such as prompt construction, prompt engineering, dataset and model loading, adversarial prompt attacks, dynamic evaluation protocols, and analysis tools.

We use the Adversarial Prompt Attacks module in PromptBench aims to evaluate the robustness of LLMs against adversarial prompts. We employ three methods to perform adversarial attacks on prompts to evaluate the adversarial robustness of the merged models: DeepWordBug (Gao et al., 2018), BERTAttack (Li et al., 2020), and StressTest (Naik et al., 2018), representing Character-level, Word-level, and Sentence-level attacks, respectively.

- **DeepWordBug** introduces subtle character-level perturbations (e.g., adding, deleting, or replacing characters) to words in text to deceive language models. It aims to evaluate a model's robustness against small typographical errors that may alter the model's performance without being easily detected.

- **BERTAttack** manipulates text at the word level by replacing words with contextually similar synonyms to mislead large language models. This method tests the model's ability to maintain accuracy despite small lexical changes that might alter the meaning of the input.

- **StressTest** appends irrelevant or redundant sentences to the end of a prompt to distract and confuse language models. It assesses the model's ability to handle extraneous information and maintain accuracy when faced with unnecessary distractions.

The evaluation is conducted on the Sentiment Analysis dataset (SST2 (Socher et al., 2013)) and the Grammar Correctness dataset (CoLA (Warstadt, 2019)):

- **SST2** (Socher et al., 2013): A sentiment analysis dataset designed to assess whether a given sentence conveys a positive or negative sentiment.

- **CoLA** (Warstadt, 2019): A dataset for grammar correctness, where the model must determine whether a sentence is grammatically acceptable.

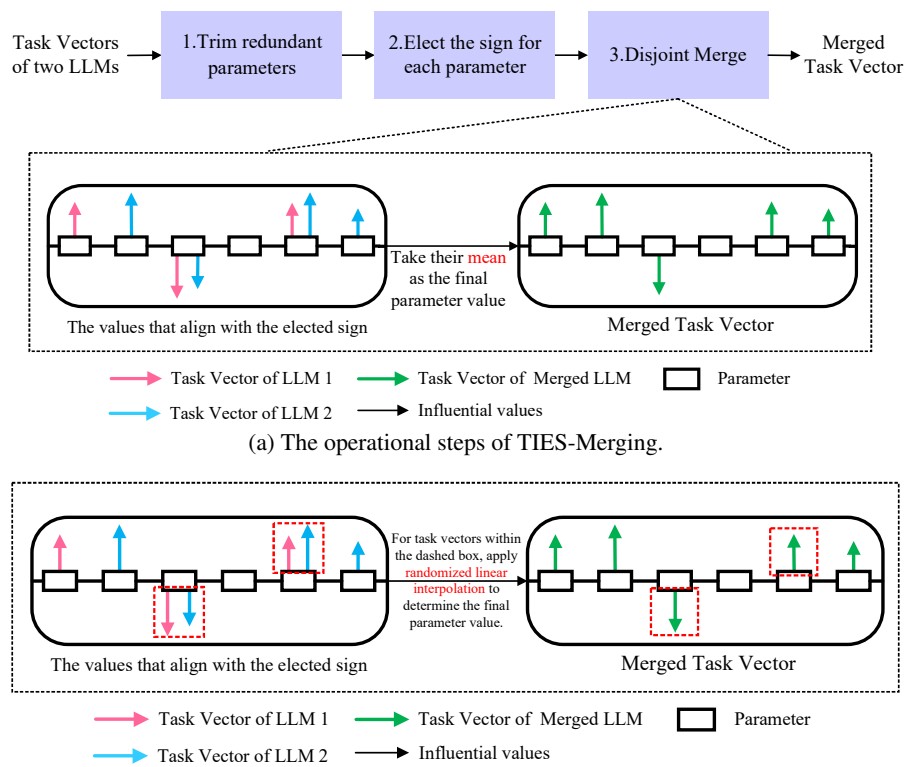

(a) The operational steps of TIES-Merging.

(b) After introducing $M^3$, the Disjoint Merge step in the TIES-Merging procedure.

Figure 5: The difference between $M^3$ and the original TIES-Merging is that, in the Disjoint Merge step, when two task vectors are retained for a given parameter, the mean of the task vectors is replaced by a random linear interpolation, while the other operations remain unchanged.

| Merging Methods | Search Ranges of Hyperparameters |
|---|---|
| Task Arithmetic | Scaling term for merging model parameters: [0.5, 0.6, 0.7, 0.8, 0.9, 1.0] |
| TIES-Merging | Scaling term for merging model parameters: [0.5, 0.6, 0.7, 0.8, 0.9, 1.0] |
| | Ratio for retaining parameters with the largest-magnitude values: [0.5, 0.7, 0.9] |

Table 6: Hyperparameter search ranges for model merging methods.

## D HYPERPARAMETER SETTING DETAILS IN MODEL MERGING METHODS

Table 6 presents the hyperparameter search ranges for the model merging methods. For Task Arithmetic and TIES-Merging, the scaling terms are selected from $[0.5, 1.0]$, while in TIES-Merging, the retain ratio for the largest-magnitude parameters is chosen from $[0.5, 0.7, 0.9]$. In contrast, the Average Merging method does not require any hyperparameters.

Table 7 presents the optimal hyperparameter settings for the TIES-Merging model merging method obtained through searching. These settings are further applied to model merging experiments involving $M^3$ and DARE.

| Merging Method | Model | Hyperparameter Values |
|---|---|---|
| | LM & Math | scaling_term=0.5, retain_ratio=0.5 |
| TIES-Merging | LM & Code | scaling_term=1.0, retain_ratio=0.7 |
| | Math & Code | scaling_term=1.0, retain_ratio=0.5 |

Table 7: Hyperparameter settings in TIES-Merging.

| Models | Merging Methods | Instruction Following | Mathematical Reasoning | Code Generating | Avg. |
|---|---|---|---|---|---|
| | | AlpacaEval | GSM8K | MBPP | |
| LM & Math | Average Merging | 45.28 | 66.34 | - | 55.81 |
| | Task Arithmetic | 45.78 | 66.34 | - | 56.06 |
| | TIES-Merging | 38.63 | 14.56 | - | 26.60 |
| | DARE | **49.00** | 66.64 | - | **57.82** |
| | DELLA | 44.03 | 48.45 | - | 46.24 |
| | M³ (Ours) | 44.90 | **67.32** | - | 56.11 |
| LM & Code | Average Merging | 36.60 | - | 32.00 | 34.3 |
| | Task Arithmetic | 44.64 | - | 33.60 | 39.12 |
| | TIES-Merging | 41.85 | - | 0.0 | 20.93 |
| | DARE | 41.47 | - | 33.00 | 37.24 |
| | DELLA | 40.99 | - | 35.00 | 38.00 |
| | M³ (Ours) | **45.20** | - | **35.20** | **40.2** |
| Math & Code | Average Merging | - | 56.17 | 8.20 | 32.19 |
| | Task Arithmetic | - | 64.67 | 8.60 | 36.64 |
| | TIES-Merging | - | 64.67 | **22.60** | **43.64** |
| | DARE | - | **65.05** | 9.80 | 37.43 |
| | DELLA | - | 63.08 | 20.2 | 41.64 |
| | M³ (Ours) | - | 64.75 | 21.4 | 43.10 |

Table 8: Comparison of our method M³ and existing model merging approaches. The best and second-best results are marked in bold and underlined fonts.

# E  COMPARISON OF M³ AND EXISTING MODEL MERGING APPROACHES

For each baseline method—Average Merging, Task Arithmetic, TIES-Merging, and DARE—we combine it with our proposed M³ method and select the best-performing combination for each merged model (LM & Math, LM & Code and Math & Code). These optimal results are reported as M³ (Ours) in Table 8.

As presented in Table 8, our proposed method M³ achieves the best overall performance across the three evaluated tasks. Importantly, M³ attains these improvements without involving complex architectural designs or intensive computations; it merely adjusts the contribution ratios of constituent models to realize substantial gains, thereby outperforming the DELLA approach. DELLA advances model sparsification techniques by building upon TIES-Merging and DARE, introducing a novel pruning strategy termed MAGPRUNE, which selects delta parameters based on their magnitudes. This strategy has inspired us to investigate new directions for enhancing the performance of merged models.

# F  PERFORMANCE DROP RATE (PDR) FOR ADVERSARIAL ROBUSTNESS

The adversarial robustness is evaluated using the Performance Drop Rate (PDR) (Zhu et al., 2023), which is defined as follows:

$$PDR = \frac{Metric_{no\ attack} - Metric_{attack}}{Metric_{no\ attack}}, \quad (1)$$

where $Metric_{no\ attack}$ denotes the performance metric without any prompt attack, and $Metric_{attack}$ represents the performance metric under the prompt attack. A smaller PDR indicates stronger adversarial defense against prompt attacks, implying better adversarial robustness.

| Model | Dataset | Use Mixup | Use Attack | Metric (%) | PDR (%) |
|---|---|---|---|---|---|
| Math & Code | SST2 | No | No | 57.68 | 11.73 |
| | | | Yes | 50.92 | |
| | | Yes | No | 78.21 | 37.10 |
| | | | Yes | 49.20 | |
| | CoLA | No | No | 72.87 | 56.97 |
| | | | Yes | 31.35 | |
| | | Yes | No | 74.02 | 58.94 |
| | | | Yes | 30.39 | |
| LM & Math | SST2 | No | No | 92.78 | 2.60 |
| | | | Yes | 90.37 | |
| | | Yes | No | 91.28 | 3.77 |
| | | | Yes | 87.84 | |
| | CoLA | No | No | 79.19 | 4.96 |
| | | | Yes | 75.26 | |
| | | Yes | No | 80.54 | 1.07 |
| | | | Yes | 79.67 | |
| LM & Code | SST2 | No | No | 10.55 | 98.91 |
| | | | Yes | 0.11 | |
| | | Yes | No | 73.17 | 97.65 |
| | | | Yes | 1.72 | |
| | CoLA | No | No | 74.21 | 8.79 |
| | | | Yes | 67.69 | |
| | | Yes | No | 73.922 | 11.15 |
| | | | Yes | 65.68 | |

Table 9: Adversarial robustness of merged models on the SST2 and CoLA datasets when executing the DeepWordBug prompt attack method.

| Model | Dataset | Use Mixup | Use Attack | Metric (%) | PDR (%) |
|---|---|---|---|---|---|
| Math & Code | SST2 | No | No | 57.68 | 13.92 |
| | | | Yes | 49.66 | |
| | | Yes | No | 78.21 | 4.11 |
| | | | Yes | 75.00 | |
| | CoLA | No | No | 45.54 | 13.47 |
| | | | Yes | 39.41 | |
| | | Yes | No | 71.72 | 17.25 |
| | | | Yes | 59.35 | |
| LM & Math | SST2 | No | No | 92.78 | 2.22 |
| | | | Yes | 90.71 | |
| | | Yes | No | 91.28 | 0.0 |
| | | | Yes | 91.28 | |
| | CoLA | No | No | 79.19 | 12.00 |
| | | | Yes | 69.70 | |
| | | Yes | No | 80.54 | 5.83 |
| | | | Yes | 75.84 | |
| LM & Code | SST2 | No | No | 10.55 | 95.65 |
| | | | Yes | 0.46 | |
| | | Yes | No | 73.17 | 55.02 |
| | | | Yes | 32.91 | |
| | CoLA | No | No | 74.21 | 7.24 |
| | | | Yes | 68.84 | |
| | | Yes | No | 73.92 | 7.52 |
| | | | Yes | 68.36 | |

Table 10: Adversarial robustness of merged models on the SST2 and CoLA datasets when executing the Bertattack prompt attack method.

## G  ADDITIONAL EXPERIMENTAL RESULTS ON ADVERSARIAL ROBUSTNESS

All the merged models are obtained using the Task Arithmetic method. Table 9 presents the detailed experimental results of the adversarial robustness of merged models on the SST2 and CoLA datasets applying the DeepWordBug prompt attack method. Table 10 presents the detailed experimental results of the adversarial robustness of merged models on the SST2 and CoLA datasets applying the BERTAttack prompt attack method.

## H  DETAILED INTRODUCTION TO DARE

DARE (Drop And REscale) (Yu et al., 2024) is a model sparsification method designed to reduce the redundancy of delta parameters in fine-tuned models while preserving their task-specific capabilities. In SFT, model parameters are optimized to unlock abilities for specific tasks, with the difference between fine-tuned and pre-trained parameters referred to as delta parameters.

However, studies have shown that delta parameters are often highly redundant. DARE addresses this redundancy by randomly dropping a proportion $p$ of delta parameters (referred to as the drop rate) and rescaling the remaining ones by a factor of $1/(1-p)$. This simple yet effective approach enables DARE to eliminate up to 99% of delta parameters with minimal impact on model performance, particularly in large-scale models, and it can be applied using only CPUs.

Beyond sparsification, DARE serves as a versatile plug-in for merging multiple homologous fine-tuned models (fine-tuned from the same base model) by reducing parameter interference. When combined with existing model merging techniques such as Average Merging, Task Arithmetic, and TIES-Merging, DARE facilitates the fusion of models while retaining or even enhancing task performance across multiple benchmarks. This effect is especially pronounced in decoder-based LMs, where DARE boosts task generalization.

Experiments on AlpacaEval, GSM8K, and MBPP reveal that the merged LM has the potential to outperform any individual source LM, presenting a significant new discovery. Notably, the 7B model obtained through DARE merging, SuperMario v2, ranks first among models of the same scale on the Open LLM Leaderboard (Beeching et al., 2023). These improvements were achieved without the need for retraining, positioning DARE as an efficient and resource-friendly solution for model merging.

## I  INTEGRATING M$^3$ INTO THE TIES-MERGING MODEL MERGING METHOD

Figure 5 shows the specific implementation approach to incorporating M$^3$ into TIES-Merging. After the steps of trimming parameters with lower magnitudes and resolving sign disagreements, the two models to be merged are denoted as $M_1$ and $M_2$. During the M$^3$ process, only the parameters that are preserved in both $M_1$ and $M_2$ are interpolated according to the model merging hyperparameter $\lambda_m$ to obtain the merged parameters. For parameters that are preserved in only one of the models, no interpolation is performed, and the original value from the preserved model is retained in the merged model.

| Merging Methods | Models | Use $M^3$ (Ours) | Use DARE | Instruction Following | Mathematical Reasoning | | Code Generating | |
|---|---|---|---|---|---|---|---|---|
| | | | | AlpacaEval | GSM8K | MATH | HumanEval | MBPP |
| Average Merging | LM & Math | Yes | No | **44.40** | 66.26 | 13.80 | - | - |
| | | No | Yes | 44.22 | **66.57** | 12.96 | - | - |
| | | Yes | Yes | 43.53 | **66.57** | **14.12** | - | - |
| | LM & Code | Yes | No | **43.91** | - | - | **37.20** | 34.40 |
| | | No | Yes | 38.81 | - | - | 31.71 | 32.40 |
| | | Yes | Yes | 40.31 | - | - | 36.59 | **37.00** |
| | Math & Code | Yes | No | - | 63.61 | 14.02 | 8.54 | 19.20↑ |
| | | No | Yes | - | 56.18 | 10.28 | 6.10 | 7.80 |
| | | Yes | Yes | - | **64.97** | 13.54 | **9.76** | **21.20** |
| Task Arithmetic | LM & Math | Yes | No | 41.65 | 66.34 | **13.74** | - | - |
| | | No | Yes | **49.00** | 66.64 | 13.02 | - | - |
| | | Yes | Yes | 44.90 | **67.32** | **13.74** | - | - |
| | LM & Code | Yes | No | **46.64** | - | - | 35.37 | 33.80 |
| | | No | Yes | 41.47 | - | - | **35.98** | 33.00 |
| | | Yes | Yes | 45.20 | - | - | **35.98** | **35.20** |
| | Math & Code | Yes | No | - | 63.53 | 13.94 | 7.93 | **19.00** |
| | | No | Yes | - | 65.05 | 13.96 | **10.37** | 9.80 |
| | | Yes | Yes | - | **65.13** | **14.32** | 8.54 | 18.00 |
| TIES-Merging | LM & Math | Yes | No | 38.73 | 18.57 | 2.48 | - | - |
| | | No | Yes | 37.92 | 18.04 | 2.34 | - | - |
| | | Yes | Yes | **39.93** | 19.26 | **2.82** | - | - |
| | LM & Code | Yes | No | 44.96 | - | - | 25.61 | 30.80 |
| | | No | Yes | 43.13 | - | - | 0.0 | 0.0 |
| | | Yes | Yes | **45.65** | - | - | **26.83** | **33.20** |
| | Math & Code | Yes | No | - | 64.75 | 14.16 | 9.76 | 21.4 |
| | | No | Yes | - | **64.82** | 13.88 | **10.37** | **23.60** |
| | | Yes | Yes | - | 64.75 | **14.78** | 9.15 | 19.60 |

Table 11: Comparison of our method $M^3$ and DARE. The best and second-best results are marked in bold and underlined fonts.

