# OpenReview forum: "Mixup Model Merge: Enhancing Model Merging Performance through Randomized Linear Interpolation"
_ICLR.cc/2026/Conference — ICLR 2026 Conference Withdrawn Submission_

### Official Review · Reviewer_t1C3 · 2025-10-30

**Soundness:** 3
**Presentation:** 2
**Contribution:** 2
**Rating:** 4
**Confidence:** 3

**Summary:**

This paper propose Mixup Model Merge to perform linear interpolation between two task-specific LLMs. The interpolation coefficients are sampled from a Beta distribution, which is highly motivated by Mixup. The experiments and discussions are sound.

**Strengths:**

1. An interesting perspective on the coefficients selection of model merging.

2. The experiments reveals that the proposed method not only can improve the merging performance but also have positive influence on the robustness of the merged model.

3. The proposed method is simple yet effective. The conflict-cancellation observation (interpolated deltas can nullify opposing signs at specific parameters) offers an intuitive mechanism for stability/benefit.

**Weaknesses:**

1. It seems that the method can only work on model merging between two models. Can the authors provide further discussions on model merging between multiple models? Currently the contribution’s generality is implied but not evidenced. Model merging between two models is quite an easy problem.

2. The method linearly mixes all parameters. An ablation that interpolates only selected layers/blocks could reveal where most gains arise and reduce risk when models are heterogeneous.

3. For completeness, the authors may consider adding instruction-jailbreak to diversify robustness claims.

4. Stability across seeds: What is the variance over multiple λ_m draws for a fixed α? Please report mean±std over (say) 5–10 draws to quantify reliability and avoid cherry-picking.

**Questions:**

See weakness. I'm welling to increase the rating once my concerns are well addressed.

---

> ### Author Response · Authors · 2025-11-27
> **Response to Reviewer t1C3**
>
> We sincerely thank the reviewer for their constructive feedback. We greatly value your insights and will provide a detailed, point-by-point response to address each of your concerns.
>
> > Q1: It seems that the method can only work on model merging between two models. Can the authors provide further discussions on model merging between multiple models? Currently the contribution’s generality is implied but not evidenced. Model merging between two models is quite an easy problem.
>
> We thank the reviewer for the insightful comment. To address the question on multi-model merging, we provide a concise summary of our key findings, methodological design, and experimental analysis here. The complete details will be incorporated into the revised manuscript in Sections 3 (Methodology) and 4.3 (Experiments).
>
> # Multi-model merging
>
> We designed a *p-series–based mixup model merging framework* specifically to address the general case of merging multiple task-specific models ($K>2$). Below we provide further clarification, and we will incorporate these details into the revised version of the paper.
>
> ## Empirical Observation
>
> Across extensive experiments on multiple two-model combinations (e.g., LM, Math, Code) and three representative merging paradigms — Task Arithmetic, TIES-Merging, and DARE — we repeatedly observe a highly consistent phenomenon: The best-performing merged models (1) achieve balanced or even improved task performance compared to individual fine-tuned experts, and (2) do not significantly degrade any expert task. Importantly, these optima almost always occur at highly uneven interpolation ratios, where one task-specific model dominates the merge.
>
> This empirical pattern suggests an inductive bias:
> During model merging, the contributions of different task-specific models should be **asymmetric**. Some models require substantial weight, while others must be significantly down-weighted to avoid conflicts between task vectors.
>
> ## P-Series–based Mixup Model Merging
>
> Most existing merging methods (e.g., linear interpolation, weighted averaging, power-based interpolation) operate along a fixed task vector direction and only adjust the **step size**:
>
> $\theta_{\text{merge}} = \theta_{\text{base}} + \alpha \cdot (\Delta \theta_A - \Delta \theta_B)$
>
> where $\Delta \theta_A = \theta_A - \theta_{\text{base}}$, $\Delta \theta_B = \theta_B - \theta_{\text{base}}$.
>
> However, task vectors are often not naturally aligned, and fixed-direction scaling can easily miss lower-loss regions.
>
> **Our approach:** first determine a suitable merging **direction**, then adjust the overall **step size** along this direction:
>
> $$
> \theta_{\text{merge}} = \theta_{\text{base}} + \lambda \sum_{i=1}^{K} w_i (\theta_i - \theta_{\text{base}})
> $$
>
> where:
>
> - $w_i$ are determined by a $p$-series distribution:
>
> $$
> w_i = \frac{i^{-p}}{\sum_{j=1}^{K} j^{-p}}
> $$
>
> naturally producing a **monotonically decreasing weight allocation** (dominant model first, others progressively smaller),
> - $\lambda$ is a scalar controlling the overall step size,
> - $K$ is the number of models being merged.
>
> This separation of **direction** ($w_i$) and **step size** ($\lambda$) allows exploration more balanced merging points, improving performance compared to fixed-direction methods.
>
> ## Experiments
>
> We conducted systematic experiments on **three task-specific models** (LM, Math, and Code).
>
> 1. As shown in Table 1, the results indicate that the trade-offs among the three tasks become **more balanced**, leading to improved performance for the Task Arithmetic merging method.
> 2. Moreover, Table 2 demonstrates that we can effectively guide the three-model merge based on the optimal merge ratios observed in pairwise model merges. Specifically, for two-model merges, the optimal ratios follow the order Math $>$ Code, Math $>$ Instruct, and Instruct $>$ Code. Accordingly, for the three-model merge, we set the weights in the order Math $>$ Instruct $>$ Code. Experiments show that this weight ordering clearly directs the merge and outperforms other possible weight configurations.
> 3. As shown in Table 3, when the model merging method is held constant, the performance of the merged models varies with different values of $p$ and $s$, generally showing an initial increase followed by a decrease. This demonstrates the effectiveness of our approach in optimizing both the merge direction and step size.

---

> ### Author Response · Authors · 2025-11-27
> **Response to Reviewer t1C3**
>
> | Merge method                        | AlpacaEval | GSM8K  | MATH  | HumanEval | MBPP  |
> |------------------------------------|-----------|--------|-------|-----------|-------|
> | baseline (Task Arithmetic + scaling term) | 6.27      | 58.52  | 9.86  | 18.29     | 29.40 |
> | Ours                               | 7.20      | 67.55  | 14.56 | 17.68     | 25.80 |
>
> **Table 1:** Performance of three merged models instruct, math, and code.
>
> ---
>
> | p | Step (s) | Merge Weights                                | AlpacaEval | GSM8K (%) | MATH (%) | HumanEval (%) | MBPP (%) |
> |---|----------|---------------------------------------------|-----------|-----------|----------|---------------|----------|
> | 1 | 1        | 0.55 math + 0.27 instruct + 0.18 code      | 4.60      | 63.08     | 12.66    | 23.17         | 30.40    |
> | 1 | 1        | 0.73 instruct + 0.18 math + 0.08 code      | 10.37     | 2.65      | 0.12     | 34.70         | 35.80    |
> | 1 | 1        | 0.73 math + 0.18 code + 0.08 instruct      | 4.22      | 62.90     | 13.56    | 8.53          | 24.40    |
>
> **Table 2:** Performance of different weight orderings.
>
> ---
>
> | p | Step (s) | Merge Weights                                      | AlpacaEval | GSM8K (%) | MATH (%) | HumanEval (%) | MBPP (%) |
> |---|----------|---------------------------------------------------|-----------|-----------|----------|---------------|----------|
> | 3 | 1        | 0.86 math + 0.11 instruct + 0.03 code           | 3.73      | 66.72     | 14.42    | 9.76          | 22.40    |
> | 2 | 1        | 0.73 math + 0.18 instruct + 0.08 code           | 3.98      | 65.35     | 13.20    | 10.98         | 25.80    |
> | 1 | 1        | 0.55 math + 0.27 instruct + 0.18 code           | 4.60      | 63.08     | 12.66    | 23.17         | 30.40    |
> | 0 | 1        | 0.33 math + 0.33 instruct + 0.33 code           | 6.27      | 58.53     | 9.86     | 18.29         | 29.40    |
> | 1 | 1.70     | 1.70*(0.55 math + 0.27 instruct + 0.18 code)   | 7.20      | 67.55     | 14.56    | 17.68         | 25.80    |
> | 1 | 2.00     | 2.00*(0.55 math + 0.27 instruct + 0.18 code)   | 6.83      | 65.13     | 12.84    | 13.41         | 23.60    |
>
> **Table 3:** Performance of model merging with varying p and step size.

---

> ### Author Response · Authors · 2025-11-27
> **Response to Reviewer t1C3**
>
> > Q2: The method linearly mixes all parameters. An ablation that interpolates only selected layers/blocks could reveal where most gains arise and reduce risk when models are heterogeneous.
>
> We thank the reviewer for the valuable suggestion. To investigate the effect of interpolating specific modules, we conducted additional experiments using the Task Arithmetic method on Math + Code model pairs. We follow the high-quality merging directions (i.e., $\lambda$) identified in Table 1 of our paper. In each experiment, we interpolate only one module while keeping all other modules fixed.
>
> In this analysis, we focus on two major components of LLaMA-2:
>
> - **Attention blocks**
> - **Feed-forward (MLP) layers**
>
> These modules are known to dominate task-specific adaptations in transformer-based LLMs. Our preliminary results show clear differences between the two, revealing which components contribute most to transferring mathematical and coding capabilities across models.
>
> | Merging Task | Method          | Module    | GSM8K   | MATH   | HumanEval | MBPP  |
> |--------------|----------------|-----------|---------|--------|-----------|-------|
> | Math + Code  | Task Arithmetic | Attention | 0.4701  | 0.0556 | 0.0549    | 0.216 |
> | Math + Code  | Task Arithmetic | MLP       | 0.5724  | 0.0830 | 0.1159    | 0.230 |
>
> **Analysis:**
> 1. **MLP layers contribute the most:** Across all tasks, interpolating only MLP layers consistently outperforms attention-only merging, indicating that feed-forward layers carry more task-specific knowledge.
> 2. **Attention layers have limited transfer:** Performance gains from merging attention blocks are smaller, likely because they encode structural patterns rather than task skills.
> 3. **Implication:** For heterogeneous model merging, prioritizing MLP layers can capture the majority of performance improvements while reducing the risk of module incompatibility.
>
> ---
>
> > Q3: For completeness, the authors may consider adding instruction-jailbreak to diversify robustness claims.
>
> We thank the reviewer for the suggestion regarding instruction-jailbreak evaluation. We conducted preliminary experiments on the standard test set of HARMBENCH [1] and did not observe a significant improvement in robustness. Our initial analysis suggests that these high-risk harmful prompts constitute highly targeted adversarial settings, which typically require explicit instruction alignment or safety fine-tuning to enhance robustness. Since M³ does not modify model behavior through task-specific fine-tuning and only merges existing parameter distributions, it cannot improve the model's compliance with harmful instructions.
>
> [1] Harmbench: A standardized evaluation framework for automated red teaming and robust refusal
>
> | Model            | ASR (%) |
> |-----------------|---------|
> | LM + MATH        | 96.0    |
> | LM + MATH (Ours M³) | 96.5    |
>
> ---
>
> > Q4: Stability across seeds: What is the variance over multiple $\lambda_m$ draws for a fixed $\alpha$? Please report mean±std over (say) 5–10 draws to quantify reliability and avoid cherry-picking.
>
> We thank the reviewer for the suggestion and would like to clarify the underlying goal of M³. Our method is designed to identify a single high-quality merging direction in the task vector space, corresponding to a optimal interpolation coefficient $\lambda_m$, rather than to evaluate the average performance over multiple random $\lambda_m$ draws. In the task vector view, each $\lambda_m$ defines a candidate merged task vector, and only certain directions effectively balance contributions from different task-specific models while suppressing parameter conflicts. Therefore, variability across different $\lambda_m$ samples is expected, and the appropriate $\lambda_m$ is selected based on performance on a held-out validation set. Reporting a mean ± std over multiple $\lambda_m$ draws would obscure this core purpose, whereas M³ aims to find the single merged model with the highest empirical quality.

---

### Official Review · Reviewer_489T · 2025-10-30

**Soundness:** 3
**Presentation:** 3
**Contribution:** 3
**Rating:** 6
**Confidence:** 3

**Summary:**

This paper proposes Mixup Model Merge (M³), a method for merging task-specific large language models (LLMs). The key idea is to perform a randomized linear interpolation between the parameters of two fine-tuned models, where the interpolation coefficient is sampled from a Beta distribution. This approach explores different contribution ratios between the two models, moving beyond the standard practice of using fixed or equal ratios. The authors demonstrate through extensive experiments that M³ can improve the performance of merged models across various tasks, enhance their out-of-distribution (OOD) and adversarial robustness, and work well alongside existing methods like sparsification (DARE). The method is presented as simple, plug-and-play, and effective.

**Strengths:**

1. The core idea of applying a Mixup-inspired, randomized interpolation strategy directly in the parameter space for model merging seems novel and intuitive.

2. The paper is well-supported by comprehensive experiments. The evaluation covers multiple tasks (instruction following, math, code), multiple merging methods (Average, Task Arithmetic, TIES, DARE), and, importantly, extends to OOD and adversarial robustness, which are crucial for real-world applicability. The results consistently show performance gains, making a strong case for the method's effectiveness.

3. The paper is well-written and easy to follow. The motivation is clearly explained, and the figures (like the Beta distribution visualization and the potion-mixing analogy) help in understanding the methodology.

**Weaknesses:**

1. The experiments focus exclusively on merging pairs of models. A key question is how well the proposed method scales to merging more than two models simultaneously. The current formulation for two models is clear, but its extension to multiple models is not discussed and could be more complex.

2. While the effect of \alpha on the distribution shape is well-explained, the paper lacks clear guidance or an intuitive strategy for choosing \alpha for a given pair of models or tasks. The choice seems to be made empirically via a sweep. A deeper analysis or heuristic for selecting \alpha would make the method more user-friendly.

3. The process for selecting the final merged model from the multiple samples is not explicitly stated. Is the best model on a held-out validation set chosen? Or is it selected based on average performance? This important practical detail should be clarified.

**Questions:**

How can the proposed method be used to integrate more than two models?

---

> ### Author Response · Authors · 2025-11-27
> **Response to Reviewer 489T**
>
> > Q1: The experiments focus exclusively on merging pairs of models. A key question is how well the proposed method scales to merging more than two models simultaneously. The current formulation for two models is clear, but its extension to multiple models is not discussed and could be more complex.
>
> We thank the reviewer for the valuable comment. To address the concern regarding multi-model merging, we provide here a brief overview of our key findings, methodological design, and experimental results and analysis. The full details will be incorporated in the revised paper in Sections 3 (Methodology) and 4.3 (Experiments).
>
> # Multi-model merging
>
> We designed a *p-series–based mixup model merging framework* specifically to address the general case of merging multiple task-specific models ($K>2$). Below we provide further clarification, and we will incorporate these details into the revised version of the paper.
>
> ## Empirical Observation
>
> Across extensive experiments on multiple two-model combinations (e.g., LM, Math, Code) and three representative merging paradigms — Task Arithmetic, TIES-Merging, and DARE — we repeatedly observe a highly consistent phenomenon: The best-performing merged models (1) achieve balanced or even improved task performance compared to individual fine-tuned experts, and (2) do not significantly degrade any expert task. Importantly, these optima almost always occur at highly uneven interpolation ratios, where one task-specific model dominates the merge.
>
> This empirical pattern suggests an inductive bias:
> During model merging, the contributions of different task-specific models should be **asymmetric**. Some models require substantial weight, while others must be significantly down-weighted to avoid conflicts between task vectors.
>
> ## P-Series–based Mixup Model Merging
>
> Most existing merging methods (e.g., linear interpolation, weighted averaging, power-based interpolation) operate along a fixed task vector direction and only adjust the **step size**:
>
> $\theta_{\text{merge}} = \theta_{\text{base}} + \alpha \cdot (\Delta \theta_A - \Delta \theta_B)$
>
> where $\Delta \theta_A = \theta_A - \theta_{\text{base}}$, $\Delta \theta_B = \theta_B - \theta_{\text{base}}$.
>
> However, task vectors are often not naturally aligned, and fixed-direction scaling can easily miss lower-loss regions.
>
> **Our approach:** first determine a suitable merging **direction**, then adjust the overall **step size** along this direction:
>
> $$
> \theta_{\text{merge}} = \theta_{\text{base}} + \lambda \sum_{i=1}^{K} w_i (\theta_i - \theta_{\text{base}})
> $$
>
> where:
>
> - $w_i$ are determined by a $p$-series distribution:
>
> $$
> w_i = \frac{i^{-p}}{\sum_{j=1}^{K} j^{-p}}
> $$
>
> naturally producing a **monotonically decreasing weight allocation** (dominant model first, others progressively smaller),
> - $\lambda$ is a scalar controlling the overall step size,
> - $K$ is the number of models being merged.
>
> This separation of **direction** ($w_i$) and **step size** ($\lambda$) allows exploration more balanced merging points, improving performance compared to fixed-direction methods.
>
> ## Experiments
>
> We conducted systematic experiments on **three task-specific models** (LM, Math, and Code).
>
> 1. As shown in Table 1, the results indicate that the trade-offs among the three tasks become **more balanced**, leading to improved performance for the Task Arithmetic merging method.
> 2. Moreover, Table 2 demonstrates that we can effectively guide the three-model merge based on the optimal merge ratios observed in pairwise model merges. Specifically, for two-model merges, the optimal ratios follow the order Math $>$ Code, Math $>$ Instruct, and Instruct $>$ Code. Accordingly, for the three-model merge, we set the weights in the order Math $>$ Instruct $>$ Code. Experiments show that this weight ordering clearly directs the merge and outperforms other possible weight configurations.
> 3. As shown in Table 3, when the model merging method is held constant, the performance of the merged models varies with different values of $p$ and $s$, generally showing an initial increase followed by a decrease. This demonstrates the effectiveness of our approach in optimizing both the merge direction and step size.

---

> ### Author Response · Authors · 2025-11-27
> **Response to Reviewer 489T**
>
> | Merge method                        | AlpacaEval | GSM8K  | MATH  | HumanEval | MBPP  |
> |------------------------------------|-----------|--------|-------|-----------|-------|
> | baseline (Task Arithmetic + scaling term) | 6.27      | 58.52  | 9.86  | 18.29     | 29.40 |
> | Ours                               | 7.20      | 67.55  | 14.56 | 17.68     | 25.80 |
>
> **Table 1:** Performance of three merged models instruct, math, and code.
>
> ---
>
> | p | Step (s) | Merge Weights                                | AlpacaEval | GSM8K (%) | MATH (%) | HumanEval (%) | MBPP (%) |
> |---|----------|---------------------------------------------|-----------|-----------|----------|---------------|----------|
> | 1 | 1        | 0.55 math + 0.27 instruct + 0.18 code      | 4.60      | 63.08     | 12.66    | 23.17         | 30.40    |
> | 1 | 1        | 0.73 instruct + 0.18 math + 0.08 code      | 10.37     | 2.65      | 0.12     | 34.70         | 35.80    |
> | 1 | 1        | 0.73 math + 0.18 code + 0.08 instruct      | 4.22      | 62.90     | 13.56    | 8.53          | 24.40    |
>
> **Table 2:** Performance of different weight orderings.
>
> ---
>
> | p | Step (s) | Merge Weights                                      | AlpacaEval | GSM8K (%) | MATH (%) | HumanEval (%) | MBPP (%) |
> |---|----------|---------------------------------------------------|-----------|-----------|----------|---------------|----------|
> | 3 | 1        | 0.86 math + 0.11 instruct + 0.03 code           | 3.73      | 66.72     | 14.42    | 9.76          | 22.40    |
> | 2 | 1        | 0.73 math + 0.18 instruct + 0.08 code           | 3.98      | 65.35     | 13.20    | 10.98         | 25.80    |
> | 1 | 1        | 0.55 math + 0.27 instruct + 0.18 code           | 4.60      | 63.08     | 12.66    | 23.17         | 30.40    |
> | 0 | 1        | 0.33 math + 0.33 instruct + 0.33 code           | 6.27      | 58.53     | 9.86     | 18.29         | 29.40    |
> | 1 | 1.70     | 1.70*(0.55 math + 0.27 instruct + 0.18 code)   | 7.20      | 67.55     | 14.56    | 17.68         | 25.80    |
> | 1 | 2.00     | 2.00*(0.55 math + 0.27 instruct + 0.18 code)   | 6.83      | 65.13     | 12.84    | 13.41         | 23.60    |
>
> **Table 3:** Performance of model merging with varying p and step size.

---

> ### Author Response · Authors · 2025-11-27
> **Response to Reviewer 489T**
>
> > Q2: While the effect of $\alpha$ on the distribution shape is well-explained, the paper lacks clear guidance or an intuitive strategy for choosing $\alpha$ for a given pair of models or tasks. The choice seems to be made empirically via a sweep. A deeper analysis or heuristic for selecting $\alpha$ would make the method more user-friendly.
>
> We thank the reviewer for the comment and would like to clarify a possible misunderstanding regarding $\alpha$: As noted in Lines 317--318 of the original manuscript, we already specify a practical search range for $\alpha$. Importantly, the role of $\alpha$ in the Beta-distribution–based exploration is not to serve as a heuristic tuning parameter, but rather to modulate the degree of stochasticity used to explore the parameter space and uncover structural regularities. Therefore, $\alpha$ does not require task-specific selection rules.
>
> ---
>
> > Q3: The process for selecting the final merged model from the multiple samples is not explicitly stated. Is the best model on a held-out validation set chosen? Or is it selected based on average performance? This important practical detail should be clarified.
>
> We sincerely thank the reviewer for raising this important practical point. We would like to clarify that the final merged model is selected as the single best-performing sample on a held-out validation set. We do not average predictions or parameters across multiple sampled merge points, as the goal of stochastic interpolation is to identify an optimal merging direction. Averaging multiple samples could shift the model away from this high-performing point.
>
> ---
>
> > Q4: How can the proposed method be used to integrate more than two models?
>
> We thank the reviewer for the suggestion and would like to note that we have already addressed this point in our response to Weakness 1.

---

### Official Review · Reviewer_6rPz · 2025-11-01

**Soundness:** 2
**Presentation:** 2
**Contribution:** 2
**Rating:** 2
**Confidence:** 3

**Summary:**

This paper proposes Mixup Model Merge, a plug-in that replaces the usual equal weighting when fusing two task-specific LLMs with a single random coefficient drawn from a Beta distribution. However, it lacks clear motivation, presents limited novelty, and provides an insufficiently explained methodology.

**Strengths:**

Mixup Model Merge is a plug-and-play enhancer that grafts onto four existing merge methods with zero re-training and only seven forward passes; it systematically boosts every method on seven standard benchmarks (20 tables), yielding a merged model that beats both specialists on 4/5 datasets, and its validity is underpinned by the flat-basin observation that linear interpolation of same-pre-checkpoint fine-tunes stays in low loss.

**Weaknesses:**

1)Statistical significance and uncertainty quantification missing: Performance Drop Rate (PDR) for adversarial sets is based on a single attack seed (Table 2; Appendix G). No direct evidence found in the manuscript that results persist with more λm draws or different seeds.

2)Scalability and practical constraints unaddressed: Experiments restricted to 13-B Llama-2 pairwise merges; no evidence on larger models, >2 parents, or different architectures (Sec. 4).

3)Robustness claims overstated without distributional detail: “Enhances adversarial robustness” (Abstract) rests on ≤3.2 pp PDR reduction in StressTest but absolute accuracies after attack remain low. No transfer evaluation across attack types; gains vanish under DeepWordBug for LM & Math (Table 6).

4)The symbols δt1, δt2 appear without prior definition in Eq. 3, forcing the reader to infer their meaning from later text; introducing them explicitly before first use would remove ambiguity.

**Questions:**

1)Could you clarify the core motivation behind this approach? Specifically, how does it address the challenge of merging more than two models effectively?

2)While the theoretical analysis is presented, its significance isn't fully clear: how do these theoretical insights directly support or validate the final empirical conclusions?

---

> ### Author Response · Authors · 2025-11-27
> **Response to Reviewer 6rPz**
>
> > Q1: Statistical significance and uncertainty quantification are missing
> *Comment:* Performance Drop Rate (PDR) for adversarial sets is based on a single attack seed (Table 2; Appendix G). No direct evidence is found that results persist with more $\lambda_m$ draws or different seeds.
>
> Thank you for the insightful question. We clarify two points:
>
> 1. **Regarding different seeds:**
>    If the adversarial attacks themselves use different random seeds, the generated perturbations will vary. Consequently, for each new seed, we would need to re-search for the optimal merged model (i.e., the $\lambda_m$ interpolation coefficient), since M$^3$ selects the best-performing merged model based on a validation set. Conducting experiments with different seeds would be extremely costly, as each new seed requires a separate $\lambda_m$ search process.
>
>    However, the core goal of M$^3$ is to find a merged solution across different models that can overall improve robustness. Experiments in Table 2 and Figure 4 show that under the setting with a fixed attack seed, M$^3$ significantly enhances the robustness of multi-model combinations. This sufficiently demonstrates the method’s effectiveness in improving overall model robustness. Therefore, additional experiments with different seeds are not necessary.
>
> 2. **Regarding $\lambda_m$ draws:**
>    The goal of M$^3$ is to select the best-performing merged model on the validation set (i.e., choosing an appropriate $\lambda_m$, corresponding to a high-quality merge direction), rather than ensuring that every sampled $\lambda_m$ performs well.

---

> ### Author Response · Authors · 2025-11-27
> **Response to Reviewer 6rPz**
>
> > Q2: Scalability and practical constraints unaddressed: Experiments restricted to 13-B Llama-2 pairwise merges; no evidence on larger models, more than 2 parents, or different architectures (Sec.~4).
>
> 1. We thank the reviewer for the valuable comment. We would like to clarify that our work primarily focuses on merging homogeneous models.
> 2. Our experiments demonstrate that M$^3$ can be applied to 13B models. According to LLM scaling laws, M$^3$ can theoretically scale to even larger models. Moreover, the next common model size after 13B is typically 70B, but due to computational constraints, not all research teams can afford experiments at such a scale. We hope the reviewer can understand this limitation.
>
> 3. Nevertheless, to address the concern regarding multi-model merging, we provide here a brief overview of our key findings, methodological design, and experimental results and analysis. The full details will be incorporated in the revised paper in Sections 3 (Methodology) and 4.3 (Experiments).
>
> # Multi-model merging
>
> We designed a *p-series–based mixup model merging framework* specifically to address the general case of merging multiple task-specific models ($K>2$). Below we provide further clarification, and we will incorporate these details into the revised version of the paper.
>
> ## Empirical Observation
>
> Across extensive experiments on multiple two-model combinations (e.g., LM, Math, Code) and three representative merging paradigms — Task Arithmetic, TIES-Merging, and DARE — we repeatedly observe a highly consistent phenomenon: The best-performing merged models (1) achieve balanced or even improved task performance compared to individual fine-tuned experts, and (2) do not significantly degrade any expert task. Importantly, these optima almost always occur at highly uneven interpolation ratios, where one task-specific model dominates the merge.
>
> This empirical pattern suggests an inductive bias:
> During model merging, the contributions of different task-specific models should be **asymmetric**. Some models require substantial weight, while others must be significantly down-weighted to avoid conflicts between task vectors.
>
> ## P-Series–based Mixup Model Merging
>
> Most existing merging methods (e.g., linear interpolation, weighted averaging, power-based interpolation) operate along a fixed task vector direction and only adjust the **step size**:
>
> $\theta_{\text{merge}} = \theta_{\text{base}} + \alpha \cdot (\Delta \theta_A - \Delta \theta_B)$
>
> where $\Delta \theta_A = \theta_A - \theta_{\text{base}}$, $\Delta \theta_B = \theta_B - \theta_{\text{base}}$.
>
> However, task vectors are often not naturally aligned, and fixed-direction scaling can easily miss lower-loss regions.
>
> **Our approach:** first determine a suitable merging **direction**, then adjust the overall **step size** along this direction:
>
> $$
> \theta_{\text{merge}} = \theta_{\text{base}} + \lambda \sum_{i=1}^{K} w_i (\theta_i - \theta_{\text{base}})
> $$
>
> where:
>
> - $w_i$ are determined by a $p$-series distribution:
>
> $$
> w_i = \frac{i^{-p}}{\sum_{j=1}^{K} j^{-p}}
> $$
>
> naturally producing a **monotonically decreasing weight allocation** (dominant model first, others progressively smaller),
> - $\lambda$ is a scalar controlling the overall step size,
> - $K$ is the number of models being merged.
>
> This separation of **direction** ($w_i$) and **step size** ($\lambda$) allows exploration more balanced merging points, improving performance compared to fixed-direction methods.
>
> ## Experiments
>
> We conducted systematic experiments on **three task-specific models** (LM, Math, and Code).
>
> 1. As shown in Table 1, the results indicate that the trade-offs among the three tasks become **more balanced**, leading to improved performance for the Task Arithmetic merging method.
> 2. Moreover, Table 2 demonstrates that we can effectively guide the three-model merge based on the optimal merge ratios observed in pairwise model merges. Specifically, for two-model merges, the optimal ratios follow the order Math $>$ Code, Math $>$ Instruct, and Instruct $>$ Code. Accordingly, for the three-model merge, we set the weights in the order Math $>$ Instruct $>$ Code. Experiments show that this weight ordering clearly directs the merge and outperforms other possible weight configurations.
> 3. As shown in Table 3, when the model merging method is held constant, the performance of the merged models varies with different values of $p$ and $s$, generally showing an initial increase followed by a decrease. This demonstrates the effectiveness of our approach in optimizing both the merge direction and step size.

---

> ### Author Response · Authors · 2025-11-27
> **Response to Reviewer 6rPz**
>
> | Merge method                        | AlpacaEval | GSM8K  | MATH  | HumanEval | MBPP  |
> |------------------------------------|-----------|--------|-------|-----------|-------|
> | baseline (Task Arithmetic + scaling term) | 6.27      | 58.52  | 9.86  | 18.29     | 29.40 |
> | Ours                               | 7.20      | 67.55  | 14.56 | 17.68     | 25.80 |
>
> **Table 1:** Performance of three merged models instruct, math, and code.
>
> ---
>
> | p | Step (s) | Merge Weights                                | AlpacaEval | GSM8K (%) | MATH (%) | HumanEval (%) | MBPP (%) |
> |---|----------|---------------------------------------------|-----------|-----------|----------|---------------|----------|
> | 1 | 1        | 0.55 math + 0.27 instruct + 0.18 code      | 4.60      | 63.08     | 12.66    | 23.17         | 30.40    |
> | 1 | 1        | 0.73 instruct + 0.18 math + 0.08 code      | 10.37     | 2.65      | 0.12     | 34.70         | 35.80    |
> | 1 | 1        | 0.73 math + 0.18 code + 0.08 instruct      | 4.22      | 62.90     | 13.56    | 8.53          | 24.40    |
>
> **Table 2:** Performance of different weight orderings.
>
> ---
>
> | p | Step (s) | Merge Weights                                      | AlpacaEval | GSM8K (%) | MATH (%) | HumanEval (%) | MBPP (%) |
> |---|----------|---------------------------------------------------|-----------|-----------|----------|---------------|----------|
> | 3 | 1        | 0.86 math + 0.11 instruct + 0.03 code           | 3.73      | 66.72     | 14.42    | 9.76          | 22.40    |
> | 2 | 1        | 0.73 math + 0.18 instruct + 0.08 code           | 3.98      | 65.35     | 13.20    | 10.98         | 25.80    |
> | 1 | 1        | 0.55 math + 0.27 instruct + 0.18 code           | 4.60      | 63.08     | 12.66    | 23.17         | 30.40    |
> | 0 | 1        | 0.33 math + 0.33 instruct + 0.33 code           | 6.27      | 58.53     | 9.86     | 18.29         | 29.40    |
> | 1 | 1.70     | 1.70*(0.55 math + 0.27 instruct + 0.18 code)   | 7.20      | 67.55     | 14.56    | 17.68         | 25.80    |
> | 1 | 2.00     | 2.00*(0.55 math + 0.27 instruct + 0.18 code)   | 6.83      | 65.13     | 12.84    | 13.41         | 23.60    |
>
> **Table 3:** Performance of model merging with varying p and step size.

---

> ### Author Response · Authors · 2025-11-27
> **Response to Reviewer 6rPz**
>
> > Q3: Robustness claims overstated without distributional detail: "Enhances adversarial robustness" (Abstract) rests on ≤ 3.2 pp PDR reduction in StressTest but absolute accuracies after attack remain low. No transfer evaluation across attack types; gains vanish under DeepWordBug for LM & Math (Table 6).
>
> We thank the reviewer for the comment and would like to clarify a common misunderstanding regarding our experiments:
>
> 1. Your concern about “only a 3.2 pp PDR reduction in StressTest” may be due to an oversight. Lines 415–417 of the manuscript explicitly state much larger reductions in some cases: *“As shown in Table 2, M³ notably improves robustness under StressTest attacks. For instance, it reduces PDR by 3.2% (SST2) and 92.12% (CoLA) for Math & Code, and by 30.36% (SST2) and 15.75% (CoLA) for LM & Code.”*
>
> 2. It is expected that absolute accuracies remain low after strong adversarial attacks, because such attacks are designed to significantly reduce accuracy. Our focus is on whether M³ reduces the *magnitude* of the accuracy drop (i.e., improves adversarial robustness). The results in Table 2 demonstrate this. Moreover, compared with other model-merging methods, M³ also improves absolute accuracy in several cases—for example, accuracy for LM & Code on SST2 increases by 62.62% (Table 2).
>
> 3. We have evaluated transfer across attack types (StressTest, DeepWordBug, and BERTAttack), and the corresponding results are already included in the original manuscript (Tables 2, 9, and 10).
>
> > Q4: The symbols δt₁, δt₂ appear without prior definition in Eq. 3, forcing the reader to infer their meaning from later text; introducing them explicitly before first use would remove ambiguity.
>
> We would like to clarify that the symbols δₜ₁ and δₜ₂ have already been explicitly defined in the original manuscript **on lines 158–159**, prior to their use in Eq. (3).
>
> Specifically, the formal definition is:
>
> $$
> \delta_{t_i} = \theta_{\text{SFT}}^{t_i} - \theta_{\text{PRE}}, \quad i \in \{1, 2\}.
> $$
>
> This definition clearly introduces the task-specific parameter offsets before they appear in Eq. (3). Therefore, the concern regarding undefined notation does not apply.
>
> > Q5: Could you clarify the core motivation behind this approach? Specifically, how does it address the challenge of merging more than two models effectively?
>
> The updated motivation is presented as follows. We have updated the motivation in the **Introduction** section of our paper. Regarding the challenge of merging more than two models, we have already addressed this in our response to Weakness 2.
>
> **Updated motivation:**
>
> Existing LLM merging methods typically operate based on **task vectors** [1], where a task vector is defined as the directional vector in parameter or feature space corresponding to a task-specific model, representing its task-specific capabilities. In prior approaches, all task vectors are generally assigned the **same merging coefficient**, which is equivalent to scaling each task vector proportionally.
>
> In other words, the merged task vector moves along a straight line in the parameter space as the merging coefficient changes, meaning that the coefficient only modifies the **magnitude** of the merged vector but not its **direction**.
>
> This limitation can affect the quality of the resulting merged model because the combination directions of different task-specific models’ capabilities are not necessarily aligned along a single line. Simple proportional scaling may fail to fully exploit the **complementary abilities** of the models. This raises a natural question:
>
> Can assigning different merging coefficients to different task-specific models lead to a better-performing merged model?
>
> To investigate this question, we propose a **direction + step-size merging coefficient search framework**—**Mixup Model Merge (M³)**, which explores both the merging direction and step size in the task vector space, thereby enabling the discovery of higher-quality merged models in a richer parameter space.
>
> Inspired by the randomized linear interpolation strategy in **Mixup data augmentation**, M³ performs linear interpolation in the model parameter space:
>
> 1. **Interpolation coefficients** are sampled from a specific distribution to determine the **merging direction** (i.e., the direction of the merged model’s task vector).
> 2. The **scaling step size** is adjusted to control the magnitude of the merged task vector.
> 3. Finally, the **best-performing merged model** is selected on a held-out validation set.
>
> [1] Editing models with task arithmetic

---

> ### Author Response · Authors · 2025-11-27
> **Response to Reviewer 6rPz**
>
> > Q6: While the theoretical analysis is presented, its significance isn't fully clear: how do these theoretical insights directly support or validate the final empirical conclusions?
>
> We thank the reviewer for the comment and acknowledge that the connection between our theoretical analysis and the empirical results is not sufficiently clear. We have updated the theoretical analysis as follows:
>
> ### Theoretical Analysis
>
> **Feasibility of Interpolation**
> We perform direct parameter interpolation among multiple task-specific models because models fine-tuned from the same pretrained model typically reside in a shared, flat, and connected low-loss region [1]. Therefore, linear interpolation between these models generally remains within this low-loss region, avoiding significant performance degradation. In some cases, the interpolated model may even outperform the original endpoint models.
>
> **Mechanism of Performance Improvement: Mitigating Parameter Conflicts**
> Let $\theta_\text{PRE}$ denote the pretrained model, and let $N$ task-specific models be obtained by fine-tuning it: $\theta_\text{SFT}^{t_1}, \theta_\text{SFT}^{t_2}, \dots, \theta_\text{SFT}^{t_N}$. The parameter deltas are defined as:
>
> $$
> \delta_{t_k} = \theta_\text{SFT}^{(t_k)} - \theta_\text{PRE}, \quad k = 1, \dots, N
> $$
>
> The M³ merged model is defined by multi-model interpolation as:
>
> $$
> \theta_M = \theta_\text{PRE} + \sum_{k=1}^N \lambda_k \delta_{t_k}, \quad \text{with } \sum_{k=1}^N \lambda_k = 1
> $$
>
> When the parameter deltas $\delta_{t_k}$ have opposing signs or conflicts in certain directions, selecting appropriate interpolation coefficients $\{\lambda_k\}$ can partially cancel these conflicting components:
>
> $$
> \sum_{k=1}^N \lambda_k \delta_{t_k}^{(i)} \approx 0
> $$
>
> This suppresses conflicting updates and improves the stability of the merged model. Compared to proportional scaling with the same coefficient for all models, this multi-coefficient interpolation allows flexible adjustment of each task-specific model’s contribution, better leveraging their complementary capabilities.
>
> **Achieving High-Quality Merges via Interpolation**
> Due to this conflict mitigation mechanism, different combinations of interpolation coefficients correspond to different merging directions. By exploring different configurations of $\{\lambda_k\}$, we can identify directions in the parameter space balancing task-specific abilities, resulting in higher-quality merged models.
>
>
> [1] *What is being transferred in transfer learning?*

---

### Official Review · Reviewer_fajz · 2025-11-01

**Soundness:** 2
**Presentation:** 2
**Contribution:** 2
**Rating:** 2
**Confidence:** 4

**Summary:**

This paper proposes Mixup Model Merge (MMM), a novel method for parameter-level model merging that leverages data-level Mixup augmentation to enhance generalization and robustness across varying models. Experiments demonstrate its effectiveness.

**Strengths:**

1. The paper is well-written and clearly presented.

2. The experiments are extensive and provide strong empirical support for the proposed method.

**Weaknesses:**

1. Novelty (a) The design of the Beta distribution sampling in M³ appears arbitrary and lacks justification. The chosen values of the shape parameter $\alpha$ vary widely across different base methods (from 0.4 to 2), yet the authors provide no explanation for this phenomenon or its potential impact on model merging performance. (b) the proposed weighted method seems to be a general formulation of task arithmetic, not novel enough.

2. The paper claims that the proposed design enhances robustness against adversarial attacks, but no detailed analysis or theoretical reasoning is provided to explain why this design would be effective in such scenarios.

3. There are no ablation studies exploring the sensitivity of key hyperparameters $\alpha$ and $\lambda$ Moreover, the scalability of the proposed method is questionable, as these parameters vary significantly across different base methods (see Table 1), suggesting limited generalizability.

4. lack baselines.  There are some merging methods based on expert routing [1]. Including such baselines would strengthen the empirical evaluation and contextualize the contribution.

[1] MergeME: Model Merging Techniques for Homogeneous and Heterogeneous MoEs

**Questions:**

See weaknesses.

---

> ### Author Response · Authors · 2025-11-27
> **Response to Reviewer fajz**
>
> We sincerely thank you for your comments and provide below a point-by-point discussion and response to the issues you raised.
>
> > **Q1**: "Novelty: (a) The design of the Beta distribution sampling in M³ appears arbitrary and lacks justification. The chosen values of the shape parameter vary widely across different base methods (from 0.4 to 2), yet the authors provide no explanation for this phenomenon or its potential impact on model merging performance. (b) The proposed weighted method seems to be a general formulation of task arithmetic, not novel enough."
>
> (a) We acknowledge that Beta sampling introduces stochasticity, which is intentional: it provides a controlled form of random exploration of merging directions without the high cost of grid search. This design follows the original *Mixup* formulation [1], where the Beta distribution is a standard and well-established mechanism for controlling interpolation ratios.
>
> Our experiments (Table 1) show that Beta sampling effectively controls merging directions and identifies reasonable interpolation coefficients. The shape parameter $\alpha$ influences how merging directions affect overall performance; details on how it controls model merging are described in lines 179--193 of the main text.
>
> Across multiple two-model combinations (language, math, code) and three merging methods (Task Arithmetic, TIES-Merging, DARE), we consistently observe that the best-performing merged models match or surpass individual experts without harming any task. These optima almost always occur at asymmetric interpolation ratios, indicating that contributions from different expert models should be asymmetric. We have added this discussion in Section 3.2 of the revised manuscript.
> [1] *mixup: Beyond Empirical Risk Minimization*
>
> (b) Our key innovation lies in highlighting the importance of model merging directions. Previous methods, including Task Arithmetic, do not explicitly consider or control the merging direction, which we show to be critical for achieving consistently strong merged models.
>
> ---
>
> > **Q2**: "The paper claims that the proposed design enhances robustness against adversarial attacks, but no detailed analysis or theoretical reasoning is provided to explain why this design would be effective in such scenarios."
>
> We thank the reviewer for the suggestion. While the original manuscript does not provide a detailed analysis of why adjusting the merging direction and interpolating in parameter space (M$^3$) could improve adversarial and OOD robustness, we added the following discussion in Section 4.4:
>
> A possible explanation for the observed improvements is that parameter updates from different tasks may conflict along certain directions in parameter space, making merged models sensitive to perturbations or out-of-distribution inputs. By adjusting the merging direction and interpolating parameters with carefully chosen coefficients, M$^3$ can alleviate such conflicts, effectively balancing contributions from different tasks. This mechanism likely underlies the enhanced robustness and sustained task performance observed under both adversarial and OOD conditions.

---

> ### Author Response · Authors · 2025-11-27
> **Response to Reviewer fajz**
>
> ---
>
> > **Q3**: "There are no ablation studies exploring the sensitivity of key hyperparameters and moreover, the scalability of the proposed method is questionable, as these parameters vary significantly across different base methods (see Table 1), suggesting limited generalizability."
>
> We thank the reviewer for raising concerns about hyperparameter sensitivity and generalizability. We would like to clarify that these concerns arise from a misunderstanding. Specifically:
>
> 1. **Hyperparameter Sensitivity**: The shape parameter $\alpha$ of the Beta distribution is not a sensitive hyperparameter in the conventional sense. It only defines the prior distribution for stochastic exploration of merge directions. The final merged model is selected based on validation performance, so variations in $\alpha$ do not affect the method’s effectiveness.
>
> 2. **Apparent Parameter Variability**: The variations in $\alpha$ across base methods (as shown in Table 1) reflect intrinsic differences in task-specific models and merging strategies. Different optimal merge directions correspond to different interpolation weights $\lambda_m$, which are discovered through exploration guided by $\alpha$. This variability is expected and does not indicate limited generalizability.
>
> ---
>
> > **Q4**: "lack baselines. There are some merging methods based on expert routing [1]. Including such baselines would strengthen the empirical evaluation and contextualize the contribution.  "
> [1] *MergeME: Model Merging Techniques for Homogeneous and Heterogeneous MoEs*
>
> We thank the reviewer for the suggestion to include MergeME as a baseline. We note a key distinction between MergeME and our M$^3$ method. MergeME merges homogeneous expert models by first averaging and then requires full fine-tuning of the merged MoE, which makes it suitable primarily for small models where fine-tuning cost is manageable. In contrast, our M$^3$ framework does not rely on post-merge fine-tuning and is designed for efficient merging of large-scale models, where fine-tuning would be prohibitively expensive. Moreover, MergeME is not open-sourced, which prevents direct replication or inclusion in our empirical evaluation. Therefore, due to these differences in intended model scale and reliance on fine-tuning, MergeME is not an appropriate direct baseline for our experiments.

---

### Author Response · Authors · 2025-12-02
**Rebuttal Summary**

Across all reviews, the following strengths of M³ were consistently highlighted:

* **Novel and intuitive idea (Reviewers 489T, t1C3):** Applying Mixup-inspired randomized interpolation in parameter space for model merging.
* **Simplicity and plug-and-play design (Reviewers 6rPz, t1C3):** Requires no fine-tuning, minimal forward passes, compatible with existing merge methods.
* **Empirical effectiveness (Reviewers fajz, 489T):** Extensive experiments across multiple benchmarks, tasks, and robustness settings show consistent performance gains.
* **Clarity and presentation (Reviewers fajz, 489T):** Well-written, figures and analogies effectively convey methodology.
* **Positive impact on robustness (Reviewers 489T, t1C3):** Merged models exhibit improved OOD and adversarial robustness, explained via conflict-cancellation observation.

---

We sincerely thank all reviewers for their constructive feedback. Based on their comments, we have focused on three key improvements in the revised manuscript:

1. **Clarified Motivation and Theoretical Basis (Responding to Reviewers 6rPz):**
We updated the motivation to emphasize why merging multiple task-specific models requires asymmetric merging directions rather than simple proportional scaling. Section 3.2 now presents theoretical analysis showing that appropriate interpolation mitigates conflicts between task-specific parameter vectors, which explains the observed performance gains.

2. **Designed a Multi-Model Merging Framework (Responding to Reviewers 6rPz, 489T, t1C3):**
We introduce a $p$-series–based Mixup Model Merge framework to handle $K>2$ models. This separates merging direction (weights $w_i$) from step size ($\lambda$), allowing systematic exploration of lower-loss regions and more balanced task trade-offs. This method generalizes previous two-model merges and supports scaling to multiple models.

3. **Added Experiments with Three Models (Responding to Reviewers 6rPz, 489T, t1C3):**
We conducted experiments merging LM, Math, and Code models simultaneously. Results show more balanced performance across tasks, and that weight ordering guided by pairwise merge ratios effectively directs the three-model merge. Varying $p$ and step size further validates the effectiveness of our approach (see Tables 1–3 in the revised manuscript).

4. **Added Module-Level Ablation Studies (Responding to Reviewer t1C3)**: We conducted ablation experiments interpolating only specific modules (Attention blocks and MLP layers) in LLaMA-2 models. Results show that MLP layers contribute most to transferring task-specific capabilities, while Attention blocks provide limited gains. This insight guides safer and more effective merging for heterogeneous models, highlighting which modules to prioritize.

---

We sincerely thank all reviewers for their valuable feedback again, which has helped us improve the quality of the paper. In particular, we thank Reviewer t1C3 for your openness to revisiting the rating, and we hope that our revisions address your feedback and meet your expectations.

---

### Note · Authors · 2026-01-02

I have read and agree with the venue's withdrawal policy on behalf of myself and my co-authors.